# Synthesis, Cytotoxic Activity Evaluation and Quantitative Structure-ActivityAnalysis of Substituted 5,8-Dihydroxy-1,4-naphthoquinones and Their *O*- and *S*-Glycoside Derivatives Tested against Neuro-2a Cancer Cells

**DOI:** 10.3390/md18120602

**Published:** 2020-11-29

**Authors:** Sergey Polonik, Galina Likhatskaya, Yuri Sabutski, Dmitry Pelageev, Vladimir Denisenko, Evgeny Pislyagin, Ekaterina Chingizova, Ekaterina Menchinskaya, Dmitry Aminin

**Affiliations:** 1G.B. Elyakov Pacific Institute of Bioorganic Chemistry of Far Eastern Branch of Russian Academy of Sciences, Prospekt 100-let Vladivostoku, 159, 690022 Vladivostok, Russia; sergpol@piboc.dvo.ru (S.P.); galin56@mail.ru (G.L.); alixar2006@gmail.com (Y.S.); pelageev@mail.ru (D.P.); vladenis@piboc.dvo.ru (V.D.); pislyagin@hotmail.com (E.P.); martyyas@mail.ru (E.C.); ekaterinamenchinskaya@gmail.com (E.M.); 2School of Natural Sciences, Far Eastern Federal University, Sukhanova St. 8, 690091 Vladivostok, Russia; 3Department of Biomedical Science and Environmental Biology, Kaohsiung Medical University, 100, Shih-Chuan 1st Road, Kaohsiung 80708, Taiwan

**Keywords:** neuroblastoma Neuro-2a cells, 5,8-dihydroxy-1,4-naphthoquinone, *O*-glucoside, thiomethylglycoside, cytotoxic activity, QSAR

## Abstract

Based on 6,7-substituted 2,5,8-trihydroxy-1,4-naphtoquinones (1,4-NQs) derived from sea urchins, five new acetyl-*O*-glucosides of NQs were prepared. A new method of conjugation of per-*O*-acetylated 1-mercaptosaccharides with 2-hydroxy-1,4-NQs through a methylene spacer was developed. Methylation of 2-hydroxy group of quinone core of acetylthiomethylglycosides by diazomethane and deacetylation of sugar moiety led to 28 new thiomethylglycosidesof 2-hydroxy- and 2-methoxy-1,4-NQs. The cytotoxic activity of starting 1,4-NQs (13 compounds) and their *O*- and *S*-glycoside derivatives (37 compounds) was determined by the MTT method against Neuro-2a mouse neuroblastoma cells. Cytotoxic compounds with EC_50_ = 2.7–87.0 μM and nontoxic compounds with EC_50_ > 100 μM were found. Acetylated *O*- and *S*-glycosides 1,4-NQs were the most potent, with EC_50_ = 2.7–16.4 μM. Methylation of the 2-OH group innaphthoquinone core led to a sharp increase in the cytotoxic activity of acetylated thioglycosidesof NQs, which was partially retained for their deacetylated derivatives. Thiomethylglycosides of 2-hydroxy-1,4-NQs with OH and MeO groups in quinone core at positions 6 and 7, resprectively formed a nontoxic set of compounds with EC_50_ > 100 μM. A quantitative structure-activity relationship (QSAR) model of cytotoxic activity of 22 1,4-NQ derivatives was constructed and tested. Descriptors related to the cytotoxic activity of new 1,4-NQ derivatives were determined. The QSAR model is good at predicting the activity of 1,4-NQ derivatives which are unused for QSAR models and nontoxic derivatives.

## 1. Introduction

Cancer is one of the leading causes of death worldwide. Brain cancer is considered one of the most insidious forms of cancer. This disease is characterized by a poor prognosis and a high rate of relapses, leading to high mortality [1]. Standard antitumor therapy procedures, including surgery, radiotherapy or chemotherapy, are oftenineffective. The removal of tumors by surgery is often impossible or difficult due to the anatomical location of the tumor and its proximity to the vital structure of the brain. Surgical excision or radiotherapy can damage these areas and disrupt the functioning of the brain. In the case of medicamentous therapy, the patient requires large doses of antitumor drugs to overcome the blood-brain barrier [2]. This, in turn, leads to an increase in the toxicity of drugs and the appearance of undesirable side effects. In this regard, considerable attention is currently being paid to the search for new antitumor compounds that can easily penetrate into the brain tissue and purposefully suppress malignant neoplasms [3,4].

Widely distributed in nature, 1,4-Napthoquinones (1,4-NQs) occur in plants, echinodermsand microorganisms [5]. The diverse activity of 1,4-NQs, from antibacterial to antitumor [6,7], makes them a promising platform inthe search of drug-leads and the designof new medicines. In various studies, 1,4-NQs and their derivatives have been tested for activity against different cancer cell lines such as colon adenocarcinoma, breast ductal carcinoma, chronic myelogenous leukemia, human cervical cancer HeLa, acute myeloid leukemia HL60, human breast cancer MCF-7 and MDA-MB-231, nonsmall cell lung cancer H1975, nasopharyngeal carcinoma HNE1, gastric cancer SG7901, human alveolar basal epithelial adenocarcinoma A549, mouse Leydig cell tumor I-10, and others, and have been shown to exhibit relatively high cytotoxic properties at micromolar concentrations [8]. Despite a fairly large number of studies on the cytotoxic effect of 1,4-NQs on various cancer cell lines aimed at elucidating the molecular mechanisms of their antitumor action, no investigations have been undertakento date on brain cancer cells.

Neuroblastoma is a common human malignant brain tumor, especially in children, that arises from the sympathetic nervous system. It is characterized by a variety of clinical features, including rapid tumor progression with long-term survival of less than 40% in spite of surgical interventions, chemotherapy, radiotherapy and biotherapy approaches [9]. The neuro-2a cell line originates from mouse brain tumor cells, and is one of the most convenient models for studying the anticancer effects of low-molecular compounds, as well as for the search for nontoxic substances to treat neurodegenerative disorders such as Alzheimer’s and Parkinson’s diseases. Neuro-2a cells are really neuronal cells that have neurites, and are even able to form some kinds of neural networks in vitro [10]. Thus, murine neuroblastoma Neuro-2a was selected as a model of human neuroblastoma and studied as part of the search for effective cytotoxic compounds to treat brain cancer.

An attractive group of natural 1,4-naphthoquinones are spinochromes, i.e., the pigments of echinoderms with naphthazarin **1** (5,8-dihydroxy-1,4-naphthoquinone) core [5,11]. Natural hydroxylated naphthazarins from echinoderms and plants are presented in Figure 1.

Hydroxynaphthazarins demonstrated various biological activities, such as antimicrobial [11], antialgae [12], cardioprotective [13,14] and antioxidant [15]. Methoxylated naphthazarins, as well as other polymethoxylated natural and semisynthetic compounds with aromatic cores, are an attractive model for the design of new anticancer agents [16]. For example, natural 2,3,6-trimethoxynaphthazarin (tricrozarin B) inhibited HeLa S_3_ cell colony formation with an IC_50_ of 0.007 μg/mL [17]. Most available sea urchin pigment echinochrome **9** is used for the treatment of ischemia, myocardial infarction, traumas and burns tothe eyes [13], as well as intraocular hemorrhages, various degenerative processes and inflammation of the eye [18]. Additionally, 5,8-dihydroxy-1,4-naphthoquinone derivatives exist in various tautomeric forms, which react with the formation of different reaction products [19] (Figure 1). The tautomeric equilibrium in the naphthazarin core depends on the nature of the substituent in core, the pH, and the reaction medium properties [20]. Recently, the ability of echinochrome to enhance mitochondrial biogenesis in cardiac [21] and skeletal muscles [22] was revealed. Echinochrome demonstrated a potential toimprove the musculoskeletal system and lipid and protein metabolism in both types of diabetes mellitus [23]. It is assumed that echinochrome activity is due to the ability of β-hydroxyl groups to inhibit radical reactions and chelatethe transition metal anions that are responsible for the initiation of free-radical oxidation in biological systems [24,25].

To improve their solubility and achieve atargeted action, naphthoquinones can easily be converted into *O*- and *S*-glycosides [26] or nonglycoside *O*- and *S*-carbohydrate conjugates with promising cytotoxic activity and selectivity [27,28,29]. Well-developed methods of chemical transformation of NQs and their biological activities have led to a number of works describing quantitative structure–activity relationship (QSAR)analyses of the cytotoxicity of NQs with various structures. The effect of substituents on the bioactivity of new NQ derivatives may be predicted using the obtained QSAR models [30,31,32,33,34,35,36].

In this study, in continuation of our drug development project, we synthesized a batch of five new NQ *O*-glycosides derived from hydroxynaphthazarins related to sea urchin pigments. The *O*-glycosidic bond attached to the quinone ring waschemically reactive and could easily be substituted under basic conditions [37], or degraded in vivo via enzyme-catalyzed hydrolysis with the release of bioactive 1,4-naphthoquinone moiety. It is known that thioglycosides are stable to acidic and basic hydrolysis and do not undergo enzymatic degradation [38]. In order to prepare NQ-sugar derivatives which were resistant to enzyme-mediated degradation, we developed a new synthetic method and synthesized thiomethyl conjugates of 1-mercaptosaccharides with NQs. In this type of naphthoquinone-thioglycoside derivative, 1-thiosugars were attached to the quinone nucleus through a methylene spacer which blocked the conjugation of sulfur atom π-orbitals with the quinone core and, therefore, did not affects its red-ox properties. The set of new NQ thiomethylglycosides and their acetyl and 2-methoxy derivatives (28 compounds) were designed and prepared. The cytotoxic activity both newly-prepared and stocked collections of NQs was determined on a model of mouse neuroblastoma Neuro-2a cancer cells. In order to evaluate the effects of chemical modifications on the cytotoxicity of a naphthazarin skeleton, a QSAR analysis was done. A QSAR model of the cytotoxic activity of 22 1,4-NQ derivatives was constructed and tested. Descriptors were determined to be related to the cytotoxic activity of the new 1,4-NQ derivatives.

## 2. Results

### 2.1. Synthesis of the O-Glucosides of Substituted Naphthazarins

In our previous research, we converted sea urchin pigments hydroxynaphthazarin **2**, spinazarin **5** and echinochrome **9** to related acetylated *mono*-, *bis*- and *trisO*-glucosides **12**, **13** and **14** by condensation with 3,4,6-tri-*O*-acetyl-α-d-glucopyranose 1,2-(*tert*-butoxy orthoacetate) **11** [39]. It was found that *tris* acetyl-*O*-glucoside of echinochrome (U-133) **14** possessed antitumor activity in vivo [40]. Treatment of acetylglucosides **12**–**14** with MeONa/MeOH led to deacetylated polar *O*-glucosides **15**–**17**, which were moderately soluble in water. It was found that the stability of glucosides **15**–**17** decreased significantly with thenumber of glucoside moieties attached to the naphthazarin core. So, if the water solution of naphthopurpurin monoglucoside **15** was stable for several days, spinazarin *bis*-*O*-glucoside **16** was stable for a day, while echinochrome *trisO*-glucoside **17** quickly decomposed after dissolution with the loss of the glucoside portions [39] (Figure 2). Taking into account the instability of deacetylated glucosides, further studies on the biological activity of stable acetyl-*O*-glycoside derivatives were performed, mainly on acetyl tris-*O*-glucoside echinochrome **14** (U-133). It was shown that U-133 exhibited pronounced antitumor activity in vivo in the model mouse Ehrlich carcinoma cells [40], and also that it hastherapeutic potential for the prevention and/or deceleration of Parkinson’s-like neurodegeneration [41,42].

### 2.2. Design and Synthesis of Simplified O-Glucoside Analogues of (U-133)

In order to find more effective derivatives derived from natural naphthazarins **3**, **4** and **6**, **7**, we created a new set of mono- and bis acetyl-*O*-glucoside derivatives **18**–**21**, which were more accessible simplified analogues of echinochrome trisglucoside (U-133) **14** (Figure 3). The synthetic 2-hydroxy-6,7-dimethylnaphthazarin **22** and its *O*-acetylglucoside derivative **23** were added to that set for comparison with isomeric quinones **4** and **19**.

The synthesis of naphthazarin *O*-glucosides **18**–**21**, **23** was carried out by autocatalytic condensation of quinones **3**, **4**, **6**, and **7** with the D-glucopyranose 1,2-(*tert*-butoxy orthoacetate **11** in dry chlorobenzene at reflux in a ratio of 1 mol of D-glucopyranose 1,2-orthoacetate **11** per one quinone β-hydroxy group as earlier for glucosides **12**, **13** and **14** [39]. In this autocatalytic variant, the activation of 1,2-orthoacetate **11** was achieved through catalysis of the acidic proton β-hydroxyl group of quinone. The reaction proceeded stereospecifically within 0.3–0.5 h and resulted in acetylated β-d-glucopyranosides **18**–**21**, **23** in yields of 60–85%.

### 2.3. Design and Synthesis of Thiomethylglycoside Derivatives of 6,7-Substituted 2-Hydroxynaphthazarines

The above-mentioned instability of hydroxynaphthazarin *O*-glucosides **15**–**17** prompted us to synthesize more stable thioglycoside conjugates with a naphthazarin core. Using the tetra-*O*-acetyl-1-mercapto-d-glucose **27** as a thiol component, and a collection of substituted naphthazarins **2**, **22**, **24**–**26** available in our laboratory, we developed a new method for thiomethylation of 6,7-substituted 2,5,8-trihydroxy-1,4-naphthoquinones with acetylthioglucose **27** and produced the corresponding acetylated thiomethylglucosides **28**–**32** in 63–85% yields with bisnaphthoquinone methanes **33**–**37** as minor byproducts (yields 8–12%) (Scheme 1).

The acetylthioglucosides **28**–**32** were readily deacetylated under treatment in MeOH/HCl solution, and resulted in polar hydrophilic thioglucosides **38**–**42** in good yields, i.e., 75–85% (Scheme 2). The methylation of 2-hydroxy derivatives **28**–**32** with diazomethane solution gave the corresponding 2-methoxyderivatives **43**–**47** in yields of 85–95%. The subsequent deacetylation 2-methoxyacetylderivatives **43**–**47** in MeOH/HCl solution led to a new set of polar 2-methoxythiomethylglucosides **48–52** (Scheme 2).

Inspired by the unique activity of echinochrome, a set of structurally-related thiomethylglycoside derivatives was formed using sea urchin pigment spinochrome D **8** (Figure 4). The free position in the core of this quinone allowed us to introduce various thiomethyl radicals with d-glucose, d-galactose, d-mannose and d-xylose moiety and obtainnew, stable lipophilic thioglycosides **56**–**59**, bearing three β-hydroxyl groups in the naphthazarin core, which were responsible for the antioxidant properties (Figure 4). The deacetylation of acetylglucoside **56** in HCl/MeOH solution proceeded with the opening of the glucose ring and the formation of an unseparated impurity of isomeric furanoside derivatives. The effective deacetylation of the thioglycosides **56**–**59** was achieved by treatment in MeONa/MeOH solution in argon atmosphere, and resulted in hydrophilic polar thiomethylglycosides **60**–**63** in good yields 59–72%.The structures of the new compounds were determined by NMR, IR spectroscopy and HR mass spectrometry. The β-configuration of the glucosidic bond was confirmed by the value of the signal of anomeric carbon C-1′ that variedbetween 82.4–83.8 ppm for acetylderivatives **56**–**59** and 85.2–86.5 ppm for glycosides **60**–**63**.

### 2.4. Cytotoxic Activity of 5,8-Dihydroxy-1,4-naphthoquinone Derivatives

The cytotoxic activity of naphthazarins from terrestrial plants **1**,**3**,**4,** sea urchin pigments **2**,**5**–**9**, their synthetic intermediates **22**, **24**–**26**, *O*-glucoside derivatives **12**–**21**, **23** and thiomethylglucosides of substituted nahthazarins **28**–**52** was assessed on mouse Neuro-2a cancer cells. The obtained results are presented in Table 1. Other thiomethylglycosidic derivatives **56**–**63** (Figure 4) derived from spinochrome D were also tested on Neuro-2a cancer cells. Both acetylated and free carbohydrate spinochrome D derivatives **56**–**63** were nontoxic at concentrations EC_50_ > 100 μM, and formed the group of new, nontoxic analogues of echinochrome **9**.

To this end, Neuro-2a cells were cultured in the presence of compounds for 24 h. Determination of cell viability in the presence of various concentrations of 5,8-dihydroxy-1,4-NQs derivatives was carried out by the MTT method followed by spectrophotometry. As a result, for each of the 50 studied compounds, the values of the half-maximum effective concentration which suppressed cell viability by 50% relative to control intact cells (EC_50_) were established (Figure 5A–F). The values of the obtained EC_50_ are presented in Table 1. Thus, for each compound, the range of cytotoxic concentrations and the EC_50_ were determined, which allowed us to further study the dependence of the biological activity of 5,8-dihydroxy-1,4-NQs derivatives on their chemical structure.

Based on the obtained EC_50_ values, all tested compounds could be conditionally divided into four groups: (a) high toxicicity (EC_50_ ≤ 5 μM); (b) moderate toxicicity (EC_50_ = 5–30 μM); (c) low toxicicity (EC_50_ = 30–90 μM); (d) nontoxic (EC_50_ > 100 μM). Nontoxic compounds accounted for the majority (56%) of all tested compounds; low toxicicity compounds comprised 14%; moderate toxicicity comprised 20% and highly toxic compounds accounted for 10% (Figure 5G, Table 1).

The results of previous studies using the QSAR method to analyze the cytotoxic activity of naphthoquinones showed that the cytotoxic activity of compounds depends on the type of cells and structural features of naphthoquinones. Thus, the effectiveness of 1,4-NQs on lung, liver and lymphocytic leukemia tumor cells depends not only on the applied concentration, but also on the type of tumor and the type of cells isolated from these tumors [8]. Currently, data on cytotoxic activity against nerve and brain cells are not available in the literature.

The aim of this study was to investigate the quantitative structure-activity relationship for naphthazarin derivative cytotoxic activity against Neuro-2a mouse neuroblastoma cells. In this study, the cytotoxic activity of 22 naphthoquinones was measured and used to build a QSAR model to determine the relationship between the naphthoquinone structure and cytotoxic activity.

#### 2.4.1. Data Set Preparation and Descriptor Calculation

In the present study, a dataset of 50 1,4-NQs derivatives (Table 1, Figure 4) was used for 3D-structure modeling and optimization with Amber10:EHT force field using the Build module of the MOE 2019.01 program. The MOE database of the energy minimized 3D-structures of 50 1,4-NQs derivatives was used for descriptor calculation with the QuaSAR module of MOE 2019.01. The EC_50_ values were converted into corresponding pEC_50_ values (−logEC_50_) to be included in the database. The pEC_50_ values determined in this work for 22 selected cytotoxic compounds with 4 < pEC_50_ < 6 were added to the database with the 3D structures of the studied compounds. The dataset of the 22 1,4-NQs derivatives with 4 < pEC_50_ < 6 was divided into training (18) and test (4) sets, which were used for the generation of a QSAR model and its validation, respectively.

#### 2.4.2. QSAR Models Generation and Validation

A QSAR analysis was performed on a data set of 22 molecules with 4 < pEC_50_ < 6. The data set was randomly divided into a training set and a test set. The training set was initially used to build the model, and the test set was used to evaluate the prediction. The 320 descriptors were calculated using the MOE QuaSAR-Model module for each molecule in the data sets. The calculated descriptors were initially screened using the QuaSAR-Contingency module of MOE, which is a statistical application designed to assist in the selection of descriptors for QSAR. Only 57 molecular descriptors were selected among the320 used descriptors after analysis of the training set (18 compounds) by the QuaSAR-Contingency module. The QuaSAR-Model module in MOE was used to generate the QSAR models with the partial least square (PLS) method. An analysis of the models using the QuaSAR-Model report made it possible to select a smaller number of the most important molecular descriptors and to obtain models with a smaller number of descriptors. The descriptors having an effect on the performance of prediction of cytotoxic activity 1,4-NQs with QSAR models and used for QSAR models generation are described in Table 2.

The QSAR models were constructed based on the 12 selected molecular descriptors using the QuaSAR-Model module in MOE 2019.01. The regression analysis of QuaSAR-Model was used to build the QSAR model using PLS for the training set (18 compounds) with a correlation coefficient (*R*2) of 0.9242 and a RMSE of 0.1285. The *Z*-score method was adopted for the detection of outliers; any compound with a *Z*-score value higher than 2.5 was considered an outlier. Compound **50** (U-625) was defined as an outlier. The QSAR model generated for the training set (17 compounds) without outlier **50** (U-625) had a correlation coefficient (*R*2) of 0.9579 and a RMSE of 0.0965 (Table 3, Figure 6). The obtained model was validated using cross-validation leave-one-out (LOO), leading to the calculation of cross-validated correlation coefficient, and used to predict the activities of the test data set in the external validation. The best QSAR model established using a training set consisting of 17 naphthoquinones and a test set of fournaphthoquinones was as follows: pEC_50_ = 5.79098(1)
+0.03973 ASA(2)
−0.01013 ASA_H(3)
−0.00227 PEOE_VSA_HYD(4)
−0.03175 PEOE_VSA_PNEG(5)
+0.02857 PEOE_VSA_POL(6)
−0.02663 PEOE_VSA_POS(7)
+0.01627 Q_VSA_HYD(8)
+0.01372 SlogP_VSA2(9)
−0.03089 SMR_VSA0(10)
+0.09770 vsa_acc(11)
+0.03843 vsa_hyd(12)
−0.05253 vsurf_S(13)
R2 = 0.95786, RMSE = 0.09650, PRED R2 = 0.9914(14)

The prediction correlation coefficient reached 0.9914, indicating that the model had better external prediction ability (Table 3, Figure 6).

An analysis of the relative importance of the 1,4-NQs QSAR model descriptors showed that the most important descriptors were the i3D descriptors vsurf_S (Interaction field surface area) and ASA (Water accessible surface area), compared to 2D descriptors (Figure 7). The activity 1,4-NQs increased with increasing of the sum of VDW surface areas of hydrophobic atoms (Å2) and of the sum of VDW surface areas (Å2) of pure hydrogen bond acceptors (not −OH). The other descriptors we used did not play such a significant role and had rather low values of relative importance of descriptors (RID) compared to ASA and vsurf_S (Figure 7). For example, in our experiments, it was found that the correlation coefficient pEC_50_ from SlogP_VSA2 amounted to R2 = 0.2353 (data not shown), and the RID value for the SlogP_VSA2 descriptor, describing the octanol/water partition coefficient of the molecules, was rather low (RID = 0.154369, Figure 7).

The QSAR model was used to predict the activities of the test data set in the external validation. An analysis of the quality of the model using the QuaSAR module of the MOE 2019.01 program showed that the experimentally determined pEC_50_ values of 5,8-dihydroxy-1,4-NQ derivatives correlated with those calculated by the program. The validation of the QSAR model showed that the QSAR model effectively predicted the activity of 1,4-NQs test set of **19** (U-443), **21** (U-421), **22** (U-434), **30** (U-518) (Table 3, Figure 6). The model correctly predicted the activities of compounds **9** (U-138) and **38** (U-635), for which pEC_50_ < 4 (Table 3).

It was found that the introduction of acetylated *O*-glucose into the structure of positions C-2, (C-2 and C-3), (C-2, C-3 and C-7) led to the appearance of a moderate cytotoxic activity (5–30 μM). An increase in cytotoxicity highly correlated with an increase in the total size of molecules and the surface area of hydrophobic atoms of these molecules. The ASA_H descriptor values describing the water accessible surface area of all hydrophobic atoms in the series of compounds **2** (U-139), **12** (U-127), **13**(U-136) and **14** (U-133) varied within the range of 148.7 < 746.9 < 1083.2 < 1519.0 Å2, which corresponded to changes in cytotoxic activity (EC_50_, μM) in the range > 100 > 16.4 > 10.6 > 8.4, respectively (Figure 8).

To elucidate the structural elements of naphthoquinone molecules (pharmacophores) which determine high cytotoxic activity, a pharmacophore analysis of active 5,8-dihydroxy-1,4-NQ derivatives and their low-active analogues was carried out. An analysis of pharmacophores showed that an important role in the toxicity of naphthoquinone compounds is played by the carbohydrate component with hydrophobic functional groups (AcO-) for both *O*- and *S-*glycosides. The introduction of a hydrophilic hydroxy group at C-2, instead of hydrophobic methoxy, reduced the cytotoxic activity, even in the presence of a hydrophobic carbohydrate component (Figure 9).

The substituents at C-2 played a key role in the manifestation of the cytotoxic activity of the naphthazarin derivatives. Derivatives with a 2-methoxy group or acetylated monosaccharide at the C-2 position exhibited cytotoxic activity, while all derivatives with a 2-hydroxy group were nontoxic (Figure 9).

## 3. Discussion

The starting naphthazarin **1** and its natural hydroxy- and methoxyderivatives **2**–**9**,**25**,**26** were nontoxic at concentrations > 100 μM, while synthetic 6,7-dimethyl- and 6,7-dichloro derivatives **22** and **24** showed moderate toxicity with EC_50_ of 82.75 ± 4.36 and 23.10 ± 1.01 μM, respectively. Conversion hydroxynaphthazarins**2–7**,**9**,**22** into respective *O*-glucosides led to cytotoxic NQ *mono, bis* and *tris* acetyl *O*-glucosides**12–14**,**18–21**, **23** with EC_50_ = 1.46–16.43 μM. Significant differences in the cytotoxicity of *mono*, *bis* and *tris* acetyl O-glycosides were not observed. The EC_50_ values varied mainly in the range of 4.46–10.60 μM, excluding acetylglucoside **12**. Deacetylation of acetyl monoglycoside **12** (EC_50_ = 16.43 ± 4.01 μM) led to polar low toxic *O*-glucoside derivative **15** (EC_50_ = 87.40 ± 2.37 μM).

The instability of hydroxynaphthazarin *O*-glucosides prompted us to synthesize stable *S*-glycoside derivatives. We developed a new method of conjugation of per-*O*-acetyl-1-mercaptosaccharides d-glucose, d-galactose, d-mannose and d-xylose with 6,7-substituted 2,5,8-trihydroxy-1,4-naphthoquinones and paraformaldehyde through the use of a methylene spacer and produced corresponding acetylated thiomethylglycosides in 63–85% yields. Acetylated thiomethylglucosides 2,5,8-trihydroxy-1,4-NQs **28–32**, **56–59** were not toxic (EC_50_ > 100 μM), excluding acetylglucosides **29** (EC_50_ = 84.00 ± 0.48 μM) and **30** (EC_50_ = 32.20 ± 4.05 μM) bearing Me groups and chlorine atoms at positions 6 and 7. After deacetylation of quinones **28**–**32** and **56**–**59**, we obtained nontoxic thiomethyl glycosides **38**–**42** and **60**–**63**, including compounds bearing methyl groups and chlorine atoms in the quinoid nucleus.The conversion of the 2-hydroxy group of the quinone core to methoxyl led to asharp increase in the cytotoxicity of acetylatedthioglucosides of 2-MeO-1,4-NQs **43**, **45**–**47** with EC_50_ = 2.72–11.61 μM, excluding acetylglucoside **44** (EC_50_ > 100 μM), which was partially retained for their deacetylated derivatives **47**–**52**. All acetylated and deacetylated 2-hydroxy-1,4-NQ thioglycoside conjugates **31**–**32**, **41**–**42**, **56**–**63** with OH and MeO groups at positions 6 and 7 of the naphthoquinone core formed a nontoxic set of compounds with EC_50_ > 100 μM.

In our previous study, a series of new, tetracyclic oxathiine-fused quinone-thioglycoside conjugates based on biologically active 1,4-naphthoquinones (chloro-, hydroxy-, and methoxysubstituted) was synthesized and characterized. These compounds showed relatively high cytotoxic activity toward various types of cancer cells such as HeLa, Neuro-2a and mouse ascites Ehrlich carcinoma, and mouse normal epithelial cell line Jb6 Cl 41-5a without pronounced selectivity for a certain type of tumor cells [29]. The positive effect of heterocyclization with mercaptosugars on cytotoxic activity for a group of 1,4-naphthoquinones was observed. The effect of chloro-, hydroxy-, and metoxysubstituents on teracycles activity was also studied, and a significant effect of the hydroxy group on activity was shown. The results of the presented work concerning the effects of substituents on biological activity are also in agreement with our early research.

A number of studies have been carried out to establish the cytotoxic activity of 1,4-NQs against various cell types and the relationship with the features of the chemical structure by the different QSAR method. Thus, it was shown that the antiproliferative and cytotoxic activity of a series of 1,4-naphthoquinone derivatives against different types of tumor cells largely depends on their hydrophobicity/polarity, partial atomic charge and total dipole moment [30,31,32,33,34,35,36].

The calculation of the molecular properties from 3D molecular fields of interaction energies is a relatively novel approach to correlate 3D molecular structures with pharmacokinetic and physicochemical properties [43]. In our investigation of 5,8-dihydroxy-1,4-naphthoquinone derivatives, the vsurf_S (Interaction field surface area) descriptor turned out to be the most important in describing the cytotoxic properties of naphthoquinones. This descriptor is included in the equation for pEC_50_ with a minus sign, which indicates that substituent increases in the interaction field, such as OH-, will lead to a decrease in pEC_50_ and a loss of toxic activity. The ASA and ASA_H descriptors for the studied naphthoquinones positively correlated with their cytotoxic activity. With an increase in the surface area of hydrophobic atoms available to water molecules, the cytotoxic activity of naphthoquinones increased. These compounds interacted more strongly with cell membranes. A comparative analysis of the descriptor valuesof highly active U-634 and nontoxic U-633 is shown in Table 4. It can be noted that the values of almost all of the model descriptors selected by QSAR decreased for U-633 compared to U-634. This change in descriptor’s values means that the nontoxic naphthoquinone derivative has low water accessible surface area, total hydrophobic and positive van der Waals surface area, octanol/water partition coefficient, i.e., hydrophobicity, molecular refractivity, surface areas of pure hydrogen bond acceptors and hydrophobic atoms, and low interaction field surface area.

It is currently known that 1,4-NQs inhibit cancer cell proliferation and growth due to: the induction of semiquinone radicals and super oxide formation followed by DNA strand breaks; the increase in intracellular ROS amount; the effect on DNA topoisomerase II, Itch protein and GPR55; and the induction of cell apoptosis and cell cycle arrest via regulation of caspase-3/7, p53, Mdm-2, Bcl-2 and Bax gene expression and MAPK, Akt and STAT3 signaling pathways [44,45,46,47,48,49,50,51,52]. To date, with the exception of GLUT1 glucose transporter, no specialized channels, pores, receptors or carriers have been found for these compounds specifically interacting with some glucose-conjugated 1,4-NQs due to the Warburg effect [28,53].

In our investigation, we showed that the cytotoxic activity of the studied 1,4-NQs is determined primarily by their hydrophobic properties. It turned out that the greater the hydrophobicity of the naphthoquinone molecule, the greater the cytotoxic effect of the compound on tumor neuronal cells. In all likelihood, it is precisely the hydrophobic properties that allow relatively small molecules of the most hydrophobic 1,4-NQs to easily penetrate biomembranes and accessthe intracellular compartment of cancer cells, where they exert a toxic effect. Probably, the compounds studied by us can penetrate biomembranes into cells along a concentration gradient due to their hydrophobicity. This is what the results of our 3D-QSAR analysis indicated. The most cytotoxic compounds were the highly hydrophobic *O*- and *S*-glycosides of 5,8-dihydroxy-1,4-naphthoquinone. Perhaps *S*-glycosides will turn out to be the preferred molecules rather than *O*-glycosides, since they are not subject to the action of various glycosidases and will retain the integrity of their chemical structure for longer in biological tissues. Theoretically, an increase in the hydrophobicity of such derivatives by replacing key substituents with more hydrophobic ones could lead to a significant increase in cytotoxic properties. In future, we plan to investigate their ability to penetrate the blood-brain barrier and enter the cells and tissues of the brain in vivo. It is possible that these properties of the selected *S*-glycosides of 1,4-NQs may be useful in the therapy of malignant neoplasm of the brain.

## 4. MaterialsandMethods

### 4.1. Chemistry

#### 4.1.1. General Chemistry (Reagents, Solvents and Equipment)

Reagents and solvents were purchased from Fluka (Taufkirchen, Germany) and Vekton (St. Peterburg, Russia). The initial quinones were purchased from Fluka or synthesized as described elsewhere. Moisture sensitive reactions were performed under calcium chloride tube protection. Chlorobenzene was preliminarily treated with concentrated sulfuric acid, washed by water, dried with calcium chloride, and distilled over phosphorous pentoxide. Acetone, benzene, ethylacetate, hexane, methanol, and toluene were distilled.The melting points were determined on a Boetius melting-point apparatus (Dresden, Germany) and are uncorrected. The ^1^H and ^13^C NMR spectra were recorded using Bruker Avance-300 (300 MHz), Bruker Avance III-500 HD (500 MHz), and Bruker Avance III-700 (700 MHz) spectrometers (Bruker Corporation, Bremen, Germany) using CDCl_3_ and DMSO-*d*_6_ as the solvents with the signal of the residual nondeuterated solvent as the internal reference. In assigning NMR spectra, 2D NMR experiments {^1^H−^1^H} COSY, {^1^H−^13^C} HMBC-qs, and {^1^H−^13^C} HSQC were used where necessary. Spin–spin coupling constants (*J*) were reported in hertz (Hz). Multiplicity was tabulated using standard abbreviations: s for singlet, d for doublet, dd for doublet of doublets, t for triplet and m for multiplet (br means broad). ESI mass spectra and ESI high resolution mass spectra were recorded on a Bruker Maxis Impact II instrument. The progress of the reaction was monitored by thin-layer chromatography (TLC) on Sorbfil plates (IMID, Krasnodar, Russia) using the following solvent systems as eluents: hexane/benzene/acetone, 3:1:1 (*v*/*v*) (System A), hexane/benzene/acetone, 2:1:1 (*v*/*v*) (System B), hexane/benzene/acetone, 2:1:2 (*v*/*v*) (System C), benzene/ethylacetate/methanol, 2:1:1 (*v*/*v*) (System D), and benzene/ethyl acetate/methanol, 7:4:2 (*v*/*v*) (System E). TLC plates were preliminary deactivated by immersion in 0.5%acetone solution of tartaric acid and drying in air. Individual substances were isolated and purified using crystallization, column chromatography, as well as preparative TLC on silica gel (Silicagel 60, 0.040–0.063 mm, Alfa Aesar, Karlsruhe, Germany). In order to reduce a residual adsorption of quinones, silica gel was preliminarily treated for 1 h with a boiling mixture of concentrated hydrochloric and nitric acids (3:1 *v*/*v*), washed with water to achievepH~ 7, and activated at 120 °C.

Naphthazarin (Fluka) (**1**) was recrystallized from ethanol, while 2-hydroxynaphthazarin (naphthopurpurin) (**2**), 2-hydroxy-3-methylnaphthazarin (**3**) and 2-hydroxy-3-ethylnaphthazarin (**4**) were prepared according with Anufriev’s paper [54];2,3-Dihydroxynaphthazarin (**5**), 2,3-dihydroxy-6-methynaphthazarin (**6**), 6-ethyl-2,3-dihydroxynaphthazarin (**7**) and spinochrome D (**8**) were obtained according the the method described in our previous papers [55,56]; 2,3,6-Trihydroxy-7-ethylnaphthazarin (echinochrome A) (**9**) was isolated from sea urchins *Scaphechinus mirabilis*; 3,4,6-Tri-*O*-acetyl-α-d-glucopyranose 1,2-(*tert*-butoxy orthoacetate) **11** was prepared according to a method described in [57]; 2-Hydroxy-6,7-dimethylnaphthazarin (**22**) and 6,7-dichloro-2-hydroxynaphthazarin (**23**) were prepared according to a method described in [14]; 2-Hydroxy-7-methoxynaphthazarin (**24**) and 2-hydroxy-6,7-dimethoxynaphthazarin (**25**) were prepared as described in our previous work [58]. Tetra-*O*-acetyl-1-thio-β-d-glucopyranose (**27**), tetra-*O*-acetyl-1-thio-β-d-galactopyranose (**53**), tetra-*O*-acetyl-1-thio-β-d-mannopyranose (**54**), tri-*O*-acetyl-1-thio-β-d-xylopyranose (**55**) were prepared as described in [59].

#### 4.1.2. General Procedure for Synthesis of the Acetylated O-Glucosides **18**–**21**, **23** by Autocatalytic Condensation of Hydroxynaphthoquinones **3**, **4**, **6**, **7** and **22** with 3,4,6-Tri-*O*-acetyl-α-d-glucopyranose 1,2-(tert-butoxy orthoacetate) **11** in Chlorobenzene (Figure 2)

Hydroxynaphthoquinone **3**, **4**, **6**, **7** or **22** (0.50 mM) and 1,2-(*tert*-butoxy orthoacetate) d-glucopyranose **11** 202 mg (0.50 mM) per one β-OH group were stirred in dry PhCl (7 mL) at reflux (20–30 min) [39]. The chlorobenzene was evaporated in vacuo and the red residue was subjected to column chromatography on silica gel, eluting with system hexane:acetone 4:1 → 2:1 *v*/*v*, to give red polar orange fraction with *R_f_*
_=_ 0.24–0.56 (A). Crystallization from hexane-acetone yielded acetylglucosides **18**, **19**, **23**. The bisglucosides **20** and **21** were isolated as amorphous red powder.

(Appendix A) 2-(Tetra-*O*-acetyl-β-d-glucopyranosyl-1-oxy)-5,8-dihydroxy-3-methylnaphthalene-1,4-dione **18** (U-444) (Appendix A). Red solid; yield 199 mg (73%); *R**_f_*= 0.32 (A); m.p. 192–193 °C. IR (CHCl_3_): 3104, 2960, 1756, 1609, 1573, 1456, 1410, 1368 см^−1^. ^1^H NMR (500 MHz, CDCl_3_): δ 1.99 (s, 3H, COCH_3_), 2.03 (s, 3H, COCH_3_), 2.04 (s, 3H, COCH_3_), 2.12 (s, 3H, COCH_3_), 2.13 (s, 3H, ArCH_3_), 3.75 (ddd, 1H, H-5′, *J* = 2.3, 5.0, 10.1 Hz), 4.10 (dd, 1H, H-6′, *J* = 2.5, 12.4 Hz), 4.19 (dd, 1H, H-6′, *J* = 5.0, 12.4 Hz), 5.15 (dd, 1H, H-4′, *J* = 9.1, 10.1 Hz), 5.27 (dd, 1H, H-2′, *J* = 7.5, 9.5 Hz), 5.32 (dd, 1H, H-3′, *J* = 9.1, 9.5 Hz), 5.69 (d, 1H, H-1′, *J* = 7.5 Hz), 7.20 (d, 1H, *J* = 6.0 Hz), 7.22 (d, 1H, *J* = 6.0 Hz), 12.26 (s, 1H, α-OH), 12.63 (s, 1H, α-OH). ^13^C NMR (125 MHz, CDCl_3_):δ 9.6 (ArCH_3_), 20.50 (COCH_3_), 20.55 (COCH_3_), 20.57 (COCH_3_), 20.74 (COCH_3_), 61.6 (C-6′), 68.3 (C-4′),71.6 (C-2′), 72.3 (C-5′), 72.5 C-3′), 99.7 (C-1′), 111.1, 111.2, 129.2, 130.4, 137.1, 153.3, 158.3, 158.9, 169.4 (COCH_3_), 169.5 (COCH_3_), 170.1 (COCH_3_), 170.4 (COCH_3_), 182.2 (C=O), 187.4 (C=O). HRMS (ESI): *m*/*z* [M + Na]^+^ calcd. for C_25_H_26_O_14_Na: 573.1209; found 573.1215.

2-(Tetra-*O*-acetyl-β-d-glucopyranosyl-1-oxy)-3-ethyl-5,8-dihydroxynaphthalene-1,4-dione **19** (U-443). Red solid; yield 239 mg (85%); *R**_f_*= 0.34 (A); m.p. 200–201 °C. IR (CHCl_3_): 1756, 1609, 1574, 1456, 1409, 1369, 1323, 1288, 1180, 1068, 1043 cm^−1^. ^1^H NMR (500 MHz, CDCl_3_):δ 1.11 (t, 3H, CH_2_CH_3_,*J* = 7.4 Hz). 1.99 (s, 3H, COCH_3_), 2.03 (s, 3H, COCH_3_), 2.04 (s, 3H, COCH_3_), 2.12 (s, 3H, COCH_3_), 2.65 (m, 2H, CH_2_CH_3_), 3.76 (ddd, 1H, H-5′, *J* = 2.5, 5.0, 10.2 Hz), 4.08 (dd, 1H, H-6′a, *J* = 2.5, 12.4 Hz), 4.17 (dd, 1H, H-6′b, *J* = 5.0, 12.4 Hz), 5.14 (dd, 1H, H-4′, *J* = 9.4, 10.2 Hz), 5.26 (dd, 1H, H-2′, *J* = 7.8, 9.4 Hz), 5.33 (dd, 1H, H-3′, *J* = 9.4 Hz), 5.80 (d, 1H, H-1′, *J* = 7.8 Hz), 7.20 (d, 1H, ArH, *J* = 9.4 Hz), 7.25 (d, 1H, ArH, *J* = 9.4 Hz), 12.26 (s, 1H, α-OH), 12.69 (s, 1H, α-OH). ^13^C NMR (125 MHz, CDCl_3_): δ 12.8 (ArCH_2_CH_3_), 17.3 (ArCH_2_CH_3_), 20.4 (COCH_3_), 20.5 (COCH_3_), 20.6 (COCH_3_), 20.7 (COCH_3_), 61.6 (C-6′), 68.3 (C-4′),71.7 (C-2′), 72.3 (C-5′), 72.5 C-3′), 99.2 (C-1′), 111.2, 111.3, 129.1, 130.4, 141.9, 152.8, 158.3, 158.8, 169.4 (COCH_3_), 169.5 (COCH_3_), 170.1 (COCH_3_), 170.4 (COCH_3_), 182.6 (C=O), 187.2 (C=O). HRMS (ESI): *m*/*z* [M + Na]^+^ calcd. for C_26_H_28_O_14_Na: 587.1351; found 587.1361.

2,3-Bis(tetra-*O*-acetyl-β-d-glucopyranosyl-1-oxy)-5,8-dihydroxy-6-methylnaphthalene-1,4-dione**20** (U-420). Red amorphous solid; 291 mg (65%); *R_f_* = 0.24 (A). IR (CHCl_3_): 1756, 1612, 1579, 1445, 1369, 1248, 1187, 1066, 1043 cm^−1^. ^1^H NMR (500 MHz, CDCl_3_):δ 2.02 (s, 3H, COCH_3_), 2.03 (s, 9H, 3 × COCH_3_), 2.04 (s, 6H, 2 × COCH_3_), 2.08 (s, 3H, COCH_3_), 2.09 (s, 3H, COCH_3_), 2.32 (s, 3H, ArCH_3_), 3.82 (m, 2H, 2H-5′), 4.11 (dd, 2H, 2H-6′, *J* = 2.7, 12.5 Hz), 4.27 (m, 1H, H-6′), 4.30 (m, 1H, H-6′), 5.24–5.32 (m, 6H, 2H-2′, 2H-3′, 2H-4′), 5.81 (d, 1H, H-1′, *J* = 6.9 Hz), 5.87 (d, 1H, H-1′, *J* = 6.9 Hz), 7.08 (s, 1H, ArH), 12.39 (s, 1H, α-OH), 12.78 (s, 1H, α-OH). ^13^C NMR (125 MHz, CDCl_3_): δ 16.4 (ArCH_3_), 20.54 (COCH_3_), 20.56 (3 × COCH_3_), 20.62 (2×COCH_3_), 20.69 (COCH_3_), 61.70 (C-6′), 61.74 (C-6′), 68.09 (C-4′), 68.15 (C-4′), 71.8 (2 × C-2′), 72.44 (C-5′), 72.49 (C-5′), 72.6 (2 × C-3′), 99.5 (C-1′), 99.6 (C-1′), 108.7, 109.4, 129.7, 142.5, 145,5, 145.8, 160.1, 160.4, 169.2 (COCH_3_), 169.3 (COCH_3_), 169.4 (2 × COCH_3_), 170.2 (2 × COCH_3_), 170.5 (2 × COCH_3_), 180.5 (C=O), 181.3 (C=O). HRMS (ESI): *m*/*z* [M + Na]^+^ calcd. for C_39_H_44_O_24_Na: 919.2108; found 919.2115.

2,3-Bis(tetra-*O*-acetyl-β-d-glucopyranosyl-1-oxy)-6-ethyl-5,8-dihydroxynaphthalene-1,4-dione **21** (U-421). Red amorphous solid; 355 mg (78%); *R_f_*=0.30 (A). IR (CHCl_3_): 17554, 1611, 1578, 1434, 1369, 1254, 1215, 1184, 1086, 1041 cm^−1^. ^1^H NMR (500 MHz, CDCl_3_):δ 1.25 (t, 3H, CH_2_CH_3_*J* = 7.5 Hz), 2.02 (s, 3H, COCH_3_), 2.03 (s, 6H, 2 × COCH_3_), 2.04 (s, 3H, COCH_3_), 2.05 (s, 6H, 2 × COCH_3_), 2.08 (s, 3H, COCH_3_), 2.09 (s, 3H, COCH_3_), 2.73 (q, 2H, ArCH_2_CH_3_, *J* = 7.5 Hz), 3.82 (m, 2H, 2H-5′), 4.11 (dd, 2H, 2H-6′, *J* = 2.5, 12.5 Hz), 4.27 (m, 1H, H-6′), 4.29 (m, 1H, H-6′), 5.22–5.32 (m, 6H, 2H-2′, 2H-3′, 2H-4′), 5.81 (d, 1H, H-1′, *J* = 7.0 Hz), 5.86 (d, 1H, H-1′, *J* = 6.8 Hz), 7.07 (s, 1H, ArH),12.43 (s, 1H, α-OH), 12.85 (s, 1H, α-OH). ^13^C NMR (125 MHz, CDCl_3_): δ 12.7 (ArCH_2_CH_3_), 20.54 (COCH_3_), 20.57 (3 × COCH_3_), 20.61 (2 × COCH_3_), 20.69 (2 × COCH_3_), 23.1 (ArCH_2_CH_3_), 61.7 (C-6′), 61.8 (C-6′), 68.1 (C-4′), 68.2 (C-4′), 71.8 (2 × C-2′), 72.4 (C-5′), 72.5 (C-5′), 72.6 (2×C-3′), 99.5 (C-1′), 99.6 (C-1′), 108.6, 109.5, 128.2, 145.5, 145,8, 148.1, 160.2, 161.0, 169.2 (COCH_3_), 169.3 (COCH_3_), 169.4 (2 × COCH_3_), 170.2 (2 × COCH_3_), 170.5 (2 × COCH_3_), 180.2 (C=O), 181.1 (C=O). HRMS (ESI): *m*/*z* [M + Na]^+^ calcd. for C_40_H_46_O_24_Na: 933.2270; found 933.2271.

2-(Tetra-*O*-acetyl-β-d-glucopyranosyl-1-oxy)-5,8-dihydroxy-6,7-dimethylnaphthalene-1,4-dione **23** (U-330). Red solid; 440 mg (78%); *R**_f_*= 0.60 (A); m.p. 221–222 °C. IR (CHCl_3_): 1758, 1604, 1456, 1416, 1375, 1279, 1166, 1070, 1037 cm^−1^. ^1^H NMR (500 MHz, CDCl_3_):δ 2.05 (s, 3H, COCH_3_), 2.06 (s, 3H, COCH_3_), 2.08 (s, 3H, COCH_3_), 2.12 (s, 3H, COCH_3_), 2.22 (s, 3H, ArCH_3_), 2.23 (s, 3H, ArCH_3_), 3.92 (ddd, 1H, H-5′, *J* = 2.4, 6.0, 8.5 Hz), 4.20 (dd, 1H, H-6′a, *J* = 3.4, 12.0 Hz), 4.26 (dd, 1H, H-6′b, *J* = 6.0, 12.0 Hz), 5.15 (m, 1H, H-4′), 5.17 (d, 1H, H-1′, *J* = 7.0 Hz), 5.33 (m, 1H, H-3′), 5.37 (m, 1H, H-2′), 6.63 (s, 1H, ArH), 12.88 (s, 1H. α-OH), 13.03 (s, 1H, α-OH). ^13^C NMR (125 MHz, CDCl_3_): δ 12.3 (ArCH_3_), 12.5 (ArCH_3_), 20.54 (COCH_3_), 20.57 (2 × COCH_3_), 20.62 (COCH_3_), 61.8 (C-6′), 68.1 (C-4′),70.6 (C-2′), 72.2 (C-3′), 72.8 C-5′), 98.6 (C-1′), 107.7, 110.7, 114.0, 141.1, 142.7, 155.6, 164.7, 169.2, 169.3 (COCH_3_), 170.1 (COCH_3_), 170.6 (COCH_3_), 172.3 (COCH_3_), 172.4 (C=O), 173.8 (C=O). HRMS (ESI): *m*/*z* [M + Na]^+^ calcd. forC_26_H_28_O_14_Na: 587.1351; found 587.1355.

#### 4.1.3. General Procedure for the Synthesis of Acetylated Thiomethylglucosides **28**–**32** by Acid-Catalytic Condensation of Hydroxynaphthoquinones **2**, **22**, **24**–**26** with Tetra-*O*-acetyl-1-thio-d-glucose **27** and Paraformaldehyde in Acetone (Scheme 1)

First, 2-Hydroxy-1,4-napthoquinones **2, 22, 24–26** (0.50 mmol) was dissolved in acetone (13 mL), to which tetra-*O*-acetyl-1-mercapto-β-d-glucopyranose **27** (273 mg, 0.75 mmol), aq HCOOH 85% (0.20 mL), and paraformaldehydepowder (90 mg, 3.00 mmol) were added. The mixture gently refluxed with mixing (2 h) until TLC indicated that the reaction was complete. The mixture was evaporated invacuowith tolueneand the solid was subjected tocolumn or preparative TLC to give two colored fractions. The polar colored fraction with *R_f_* = 0.39–0.65 was tetra-*O*-acetyl-β-d-glucopyranosyltiomethyl conjugate **28**–**32**, and the second colored fraction with *R_f_* = 0.82–0.92 was 3,3′-bis(2-hydroxynaphthalene-1,4-dione)methane **33**–**37** (Scheme 1).

3-(Tetra-*O*-acetyl-β-d-glucopyranosyl-1-thiomethyl)-2,5,8-trihydroxynaphthalene-1,4-dione **28** (U-633).Red solid; 234 mg (80%); *R_f_* = 0.43 (B); m.p. 152–154 °C. IR (CHCl_3_): 3400, 1755, 1637, 1606, 1571, 1458, 1414, 1375, 1329, 1240, 1193, 1040 cm^−1^. ^1^H NMR ^1^H (500 MHz, CDCl_3_): δ 1.99 (s, 3H, COCH_3_), 2.00 (s, 3H, COCH_3_), 2.02 (s, 3H, COCH_3_), 2.03 (s, 3H, COCH_3_), 3.67 (m, 1H, H-5), 3.74 (d, 1H, CH_2_-S, *J* = 13.7 Hz), 3.93 (d, 1H, CH_2_-S, *J* = 13.7 Hz), 3.97 (dd, 1H, H-6′, *J* = 2.8, 4.6, 12.3 Hz), 4.18 (dd, 1H, H-6′, *J* = 4.6, 12.3 Hz), 4.68 (d, 1H, H-1′, *J* = 10.7 Hz), 5.05 (m, 1H, H-2′), 5.08 (m, 1H, H-4′), 5.21 (m, 1H, H-3′), 7.22 (d, 1H, ArH, *J* = 9.3 Hz), 7.32 (d, 1H, ArH, *J* = 9.3 Hz), 7.67 (s, 1H, β-OH), 11.48 (s, 1H, α-OH), 12.69 (s, 1H, α-OH). ^13^C NMR (125 MHz, CDCl_3_): δ 20.55 (COCH_3_), 20.58 (COCH_3_), 20.65 (COCH_3_), 20.66 (COCH_3_), 21.7 (CH_2_S), 62.1 (C-6′), 68.5 (C-4′), 70.0 (C-2′), 73.9 (C-3′), 75.8 (C-5′), 83.9 (C-1′), 110.3, 110.5, 122.4, 128.0, 132.0, 153.8, 157.4, 158.1, 169.3(COCH_3_), 169.4(COCH_3_), 170.2(COCH_3_), 170.6(COCH_3_), 181.8 (C=O), 187.4 (C=O). HRMS (ESI): *m*/*z* [M − H]^−^ calcd. for C_25_H_25_O_14_S:581.0971; found 581.0967.

3-(Tetra-*O*-acetyl-β-d-glucopyranosyl-1-thiomethyl)-2,5,8-trihydroxy-6,7-dimethylnaphthalene-1,4-dione **29** (U-519). Red solid;244 mg(82%). *R_f_* = 0.40 (B); m.p. 104–106 °C. IR (CHCl_3_): 3395, 3023, 2954, 1755, 1624, 1598, 1450, 1394, 1376, 1335, 1249, 1192, 1096, 1038 cm^−1^. ^1^H NMR ^1^H (500 MHz, CDCl_3_): δ 1.99 (s, 6H, 2 × COCH_3_), 2.01 (s, 3H, COCH_3_), 2.03 (s, 3H, COCH_3_), 2.27 (s, 3H, ArCH_3_), 2.30 (s, 3H, ArCH_3_), 3.67 (m, 1H, H-5′), 3.74 (d, 1H, CH_2_-S, *J* = 13.4 Hz), 3.93 (d, 1H, CH_2_-S, *J* = 13.4 Hz), 3.96 (dd, 1H, H-6′, *J* = 2.4, 12.0 Hz), 4.19 (dd, 1H, H-6′, *J* = 4.3, 12.0 Hz), 4.70 (d, 1H, H-1′, *J* = 10.0 Hz), 5.04 (m, 1H, H-2′), 5.08 (m, 1H, H-4′), 5.21 (m, 1H, H-3′), 7.68 (s, 1H, β-OH), 12.15 (s, 1H, α-OH), 13.42 (s, 1H, α-OH). ^13^C NMR ^1^H (125 MHz, CDCl_3_): δ 12.15 (CH_3_), 12.8 (CH_3_), 20.5 (COCH_3_), 20.6 (COCH_3_), 20.7 (COCH_3_), 21.9, 62.1 (C-6′), 68.5 (C-4′), 70.1 (C-2′), 74.0 (C-3′), 75.8 (C-5′), 84.0 C-1′), 107.5, 107.8, 121.7, 136.6, 141.5, 153.9, 157.8, 158.6, 169.3 (COCH_3_), 169.4(COCH_3_), 170.2(COCH_3_), 170.6 (COCH_3_), 180.1 (C=O), 186.4 (C=O). HRMS (ESI): *m*/*z* [M −H]^−^ calcd. for C_27_H_29_O_14_S: 609.1284; found 609.1286.

3-(Tetra-*O*-acetyl-β-d-glucopyranosyl-1-thiomethyl)-6,7-dichloro-2,5,8-trihydroxynaphthalene-1,4-dione **30** (U-518). Red solid; 224 mg (69%); *R_f_* = 0.39 (B); m.p. 106–108 °C. IR (CHCl_3_): 3407, 3021, 2956, 2360, 1755, 1630, 1610, 1553, 1432, 1402, 1379, 1323, 1246, 1221, 1182, 1117, 1040 cm^−1^. ^1^H NMR ^1^H (300 MHz, CDCl_3_): δ 2.01 (s, 6H, 2 × COCH_3_), 2.03 (s, 3H, COCH_3_), 2.04 (s, 3H, COCH_3_), 3.67 (м, 1H, H-5′), 3.76 (d, 1H, CH_2_-S, *J* = 14.0 Hz), 3.93 (d, 1H, CH_2_-S, *J* = 14.0 Hz), 4.01 (dd, 1H, H-6′a, *J* = 3.0, 12.5 Hz), 4.19 (dd, 1H, H = 6′b, *J* = 4.3, 12.5 Hz), 4.65 (d, 1H, H-1′, *J* = 10.3 Hz), 5.06 (m, 1H, H-2′), 5.08 (m, 1H, H-4′), 5.22 (m, 1H, H-3′), 7.95 (s, 1H, β-OH), 12.06 (s, 1H, α-OH), 13.37 (s, 1H, α-OH). ^13^C NMR ^1^H (75 MHz, CDCl_3_): δ 20.5 (2 × COCH_3_), 20.58 (COCH_3_), 20.60 (COCH_3_), 21.4 (CH_2_S), 61.9 (C-6′), 68.4 (C-2′), 69.8 (C-2′), 73.6 (C-3′), 75.7 (C-5′), 83.6(C-1′), 108.8 (2), 122.6, 131.5, 135.3, 153.8, 154.4, 154.5, 169.3 (COCH_3_), 169.4 (COCH_3_), 170.1 (COCH_3_), 170.5 (COCH_3_), 180.9 (C=O), 186.5 (C=O). HRMS (ESI): *m*/*z* [M − H]^−^ calcd. for C_25_H_23_HCl_2_O_14_S: 649.0191; found 649.0191.

3-(Tetra-*O*-acetyl-β-d-glucopyranosyl-1-thiomethyl)-2,5,8-trihydroxy-7-methoxynaphthalene-1,4-dione **31** (U-639). Red solid; 220 mg (72%); *R_f_* = 0.53 (C), m.p. 117–120 °C. IR (CHCl_3_): 3510, 3397, 3085, 2948, 1755, 1599, 1466, 1415, 1403, 1376, 1339, 1298, 1258, 1218, 1180, 1137, 1040 cm^−1^. ^1^H NMR (500 MHz, CDCl_3_):δ1.997 (s, 3H, COCH_3_), 2.00 (s, 3H, COCH_3_), 2.02 (s, 3H, COCH_3_), 2.04 (s, 3H, COCH_3_), 3.67 (ddd, 1H, H-5′, *J* = 9.5, 4.5, 2.7 Hz), 3.76 (d, 1H, CH_2_S,*J* = 13.5 Hz), 3.95 (d, 1H, CH_2_S,*J* = 13.5 Hz), 3.97 (dd, 1H, H-6′a, *J* = 12.3, 2.7 Hz), 3.98 (s, 3H, OCH_3_), 4.20 (dd, 1H, H-6′b, *J* = 12.7, 4.5 Hz), 4.72 (d,1H, H-1′*J* = 10.2 Hz), 5.05 (dd,1 -2′,*J* = 10.2, 9.5 Hz), 5.08 (t, 1H, H-4′, *J* = 9.5 Hz), 5.21 (t, 1H, H-3′, *J* = 9.5 Hz), 6.59 (s, 1H, H-6), 7.61 (br s, 1H, β-OH), 12.03 (s, 1H, α-OH), 13.19 (s, 1H, α-OH). ^13^C NMR (125 MHz, CDCl_3_):δ 20.5 (2 × COCH_3_), 20.6 (2 × COCH_3_), 21.8, 56.7 (OCH_3_), 62.0 (C-6′), 68.5 (C-4′), 70.0 (C-2′), 73.9 (C-3′), 75.7 (C-5′), 83.9 (C-1′), 103.9, 108.6, 110.3, 123.2, 153.1, 154.9, 157.0, 163.4, 169.3 (2 × COCH_3_), 170.2 (COCH_3_), 170.6 (COCH_3_), 177.7 (C=O), 181.9 (C=O). HRMS (ESI): *m*/*z* [M − H]^−^ calcd. for C_26_H_27_O_15_S: 611.1076; found: 611.1077.

3-(Tetra-*O*-acetyl-β-d-glucopyranosyl-1-thiomethyl)-2,5,8-trihydroxy-6,7-dimethoxynaphthalene-1,4-dione **32** (U-637). Red solid; 244 mg (76%); *R_f_* = 0.60(C), m.p. 83–85 °C. IR (CHCl_3_): 3410, 3083, 2944, 1755, 1601, 1478, 1435, 1412, 1396, 1367, 1324, 1304, 1273, 1254, 1236, 1179, 1150, 1040 cm^−^^1^. ^1^H NMR (500 MHz, CDCl_3_):δ1.99 (s, 3H, COCH_3_), 2.00 (s, 3H, COCH_3_), 2.02 (s, 3H, COCH_3_), 2.04 (s, 3H, COCH_3_), 3.66 (ddd, 1H, H-5′, *J* = 9.5, 4.5, 2.7 Hz), 3.75 (d, 1H, CH_2_S, *J* = 13.5 Hz), 3.95 (d,1H, CH_2_S, *J* = 13.5 Hz), 3.99 (dd, 1H, H-6′a, *J* = 12.3, 2.7 Hz), 4.06 (s, 3H, OCH_3_), 4.15 (s, 3H, OCH_3_), 4.18 (dd, 1H, H-6′b, *J* = 12.7, 4.5 Hz), 4.68 (d,1H, H-1′, *J* = 10.2 Hz), 5.04 (dd, 1H, H-2′, *J* = 10.2, 9.5 Hz), 5.08 (t, 1H, H-4′, *J* = 9.5 Hz), 5.21 (t, 1H, H-3′, *J* = 9.5 Hz), 7.62 (br s, 1H, β-OH), 12.15 (s, 1H,α-OH), 13.34 (s, 1H, α-OH). ^13^C NMR (125 MHz, CDCl_3_):δ 20.5 (2 × COCH_3_), 20.6 (2 × COCH_3_), 21.8, 61.6 (OCH_3_), 61.7 (OCH_3_), 62.1 (C-6′), 68.5 (C-4′), 70.0 (C-2′), 73.9 (C-3′), 75.7 (C-5′), 83.9 (C-1′), 106.1, 106.8, 121.7, 146.4, 150.4, 153.8, 159.6, 160.8, 169.3 (COCH_3_), 169.4 (COCH_3_), 170.2 (COCH_3_), 170.6 (COCH_3_), 173.9 (C=O), 181.5 (C=O). HRMS (ESI): *m*/*z* [M − H]^−^ calcd. for C_27_H_29_O_16_S: 641.1182; found: 641.1182.

#### 4.1.4. General Procedure for Synthesis of Thiomethylglucosides **38**–**42** and **48**–**52** by Acid-Catalytic Deacetylation Acetylthiomethylglucosides **28**–**32** and **43**–**47** in Methanol (Scheme 2)

To a partially dissolved suspension of (tetra-*O*-acetyl-β-d-glucopyranosyl-1-thiomethyl)naphthalene-1,4-dione **28**–**32** or **43**–**47** (0.20mmol) in dry methanol (15 mL) was added acetylchloride (1.0 mL) dropwise under vigorous stirring; then, the flask was carefully closed. The obtained reaction mixture was stirred for 48 h at room temperature, and then toluene (15 mL) was added; the resulting red solution evaporated under reduced pressure. The residue was subjected topreparative TLC (system B), yielding a polar red colored solid. The solid was crystallized with MeOH to give **38**–**42** or **48**–**52** as red crystals.

3-(β-d-Glucopyranosyl-1-thiomethyl)-2,5,8-trihydroxynaphthalene-1,4-dione **38** (U-635). Red solid; 71 mg (86%); *R_f_* = 0.26 (D); m.p. 169–170 °C. IR (KBr): 3422, 1600, 1561, 1453, 1410, 1385, 1296, 1181, 1108, 1077, 1031, 970 cm^−1^. ^1^H NMR ^1^H (500 MHz, DMSO-*d*_6_): δ 2.96 (m, 1H, H-2′), 3.02 (m, 1H, H-5′), 3.10 (m, 2H, H-3′, H-4′), 3.40 (dd, 1H, H-6a′, *J* = 5.1, 11.9 Hz), 3.50 (dd, 1H, H-6b′, *J* = 2.1, 11.9 Hz), 3.63 (d, 1H, CH_2_-S-, *J* = 13.2 Hz), 3.75 (d, 1H, CH_2_-S-, *J* = 13.2 Hz), 4.38 (d, 1H, H-1′, *J* = 9.7 Hz), 7.31 (d, 1H, ArH, *J* = 9.5 Hz), 7.37 (d, 1H, ArH, *J* = 9.5 Hz), 11.71 (s, 1H, α-OH), 12.79 (s, 1H, α-OH). ^13^C NMR ^1^H (125 MHz, DMSO-*d*_6_): δ 21.2 (CH_2_-S), 60.8 (C-6′), 69.6 (C-4′), 73.0 (C-2′), 78.4 (C-3′), 80.7 (C-5′), 86.1 (C-1′), 110.8, 111.1, 122.7, 127.6, 130.2, 155.5, 156.2, 156.6, 182.9 (C=O), 188.0 (C=O). HRMS (ESI): *m*/*z* [M − H]^−^ calcd. for C_17_H_18_O_10_S: 413.0548; found 413.0551.

3-(β-d-Glucopyranosyl-1-thiomethyl)-2,5,8-trihydroxy-6,7-dimethylnaphthalene-1,4-dione **39** (U-520). Red solid; 84 mg (95%); *R_f_* = 0.44 (D); m.p. 179–181 °C. IR (KBr): 3426, 2920, 2360, 1592, 1423, 1384, 1312, 1287, 1183, 1149, 1089, 1031 cm^−1^. ^1^H NMR ^1^H (500 MHz, DMSO-*d*_6_): δ 2.21 (s, 3H, ArCH_3_), 2.22 (s, 3H, ArCH_3_), 2.95 (m, 1H, H-2′), 3.02 (m, 1H, H-5′), 3.10 (m, 1H, H-3′), 3.13 (m, 1H, H-4′), 3.40 (dd, 1H, H-6a′, *J* = 5.0, 12.2 Hz), 3.50 (dd, 1H, H-6b′, *J* = 2.4, 12.2 Hz), 3.63 (d, 1H, CH_2_-S, *J* = 13.0 Hz), 3.75 (d, 1H, CH_2_-S, *J* = 13.0 Hz), 4.37 (d, 1H, H-1′, *J* = 10.0 Hz), 12.52 (s, 1H, α-OH), 13.53 (s, 1H, α-OH). ^13^C NMR ^1^H (125 MHz, DMSO-*d*_6_): δ 11.9 (ArCH_3_), 12.4 (ArCH_3_), 21.3 (CH_2_-S), 60.8 (C-6′), 69.6(C-4′), 73.1 (C-2′), 78.5 (C-3′), 80.7 (C-5′), 86.1 (C-1′), 107.5, 108.2, 122.6, 135.7, 1 39.3, 156.2, 156.4, 157.2, 181.3 (C=O), 186.5 (C=O). HRMS (ESI): *m*/*z* [M −H]^−^calcd. for C_19_H_22_O_10_S: 441.0861; found 441.0864.

6,7-Dichloro-3-(β-d-glucopyranosyl-1-thiomethyl)-2,5,8-trihydroxynaphthalene-1,4-dione **40** (U-624). Red solid 83 mg (86%); *R_f_* = 0.38 (D); m.p. 184–186 °C. IR (KBr): 3433, 2361, 1597, 1549, 1385, 1298, 1270, 1178, 1114, 1034, 996 cm^−1^. ^1^H NMR ^1^H (500 MHz, DMSO-*d*_6_): δ 2.96 (m, 1H, H-2′), 3.03 (m, 1H, H-5′), 3.09 (m, 2H, H-3′, H-4′), 3.37 (dd, 1H, H-6′a, *J* = 5.3, 11.9 Hz), 3.52 (dd, 1H, H-6′b, *J* = 2.1, 11.9 Hz), 3.65 (d, 1H, CH_2_-S, *J* = 12.9 Hz), 3.74 (d, 1H, CH_2_-S, *J* = 12.9 Hz), 4.38 (d, 1H, H-1′, *J* = 9.8 Hz), 12.30 (s, 1H, α-OH), 14.04 (s, 1H, α-OH). ^13^C NMR ^1^H (125 MHz, DMSO-*d*_6_): δ 21.5 (CH_2_S), 60.9 (C-6′), 69.8 (C-4′), 73.0 (C-2′), 78,4 (C-3′), 80.7 (C-5′), 86.1 (C-1′), 110.3, 110.8, 121.5, 128.5, 131.4, 152.4, 152.5, 159.0, 182.6(C=O), 185.8 (C=O). HRMS (ESI): *m*/*z* [M −H]^−^calcd. for C_17_H_15_Cl_2_O_10_S: 480.9768; found 480.9770.

3-(β-d-Glucopyranosyl-1-thiomethyl)-2,5,8-trihydroxy-7-methoxynaphthalene-1,4-dione **41** (U-644). Red solid; 61 mg (69%); *R_f_* = 0.38 (E); m.p. 211–213 °C. IR (KBr):3412, 2360, 1590, 1477, 1424, 1385, 1312, 1200, 1164, 1111, 1064, 1029, 984, 953 cm^−^^1^.^1^H NMR (700 MHz, DMSO-*d*_6_):δ2.95 (dd, 1H, H-2′, *J* = 9.8, 8.5 Hz), 3.03 (ddd,1H, H-5′,*J* = 8.5, 5.0, 2.0 Hz), 3.10 (t, 1H, H-3′, *J* = 8.5 Hz), 3.13 (t, 1H, H-4′, *J* = 8.5 Hz), 3.41 (dd, 1H, H-6′a, *J* = 11.8, 5.0 Hz), 3.51 (dd, 1H, H-6′b, *J* = 11.8, 2.0 Hz), 3.63 (d, 1H, ArCH_2_, *J* = 13.0 Hz), 3.77 (d, 1H, ArCH_2_, *J* = 13.0 Hz), 3.91 (s, 3H, OCH_3_), 4.39 (d,1H, H-1′,*J* = 9.8 Hz), 6.77 (s, 1H, H-6), 11.42 (br s, 1H, β-OH), 12.22 (s, 1H, α-OH), 13.37 (s, 1H, α-OH). ^13^C NMR (175 MHz, DMSO-*d*_6_):δ 21.3(CH_2_S), 56.8 (OCH_3_), 60.8 (C-6′), 69.7 (C-4′), 73.1 (C-2′), 78.4 (C-3′), 80.7 (C-5′), 86.0 (C-1′), 103.4, 108.3, 110.6, 123.4, 155.1, 155.2, 156.9, 163.4, 176.9 (C=O), 181.0 (C=O). HRMS (ESI): *m*/*z* [M − H]^−^ calcd. for C_18_H_19_O_11_S: 443.0654; found: 443.0654.

3-(β-d-Glucopyranosyl-1-thiomethyl)-2,5,8-trihydroxy-6,7-dimethoxynaphthalene-1,4-dione **42** (U-640). Red solid; 71 mg (75%); *R_f_* = 0.42 (E); m.p. 193–195 °C. IR (KBr):3354, 2957, 2854, 1604, 1461, 1423, 1384, 1330, 1276, 1215, 1180, 1161, 1123, 1100, 1060, 1029, 986 cm^−^^1^. ^1^H NMR (500 MHz, DMSO-*d*_6_):δ2.95 (dd, 1H, H-2′, *J* = 9.8, 8.5 Hz), 3.03 (ddd, 1H, H-5′, *J* = 8.5, 5.0, 2.0 Hz), 3.09 (t, 1H, H-3′, *J* = 8.5 Hz), 3.13 (t, 1H, H-4′, *J* = 8.5 Hz), 3.42 (dd,1H, H-6′a,*J* = 11.8, 5.0 Hz), 3.53 (dd, 1H, H-6′b*, J* = 11.8, 2.0 Hz), 3.64 (d, 1H, ArCH_2_,*J* = 13.0 Hz), 3.79 (d, 1H ArCH_2_S, *J* = 13.0 Hz), 3.96 (s, 3H, OCH_3_), 4.01 (s, 3H, OCH_3_), 4.37 (d, 1H, H-1′, *J* = 9.8 Hz), 11.47 (br s, 1H, β-OH), 12.43 (s, 1H, α-OH), 13.43 (s, 1H, α-OH).^13^C NMR (125 MHz, DMSO-*d*_6_):δ 21.2, 60.8 (C-6′), 61.3 (2 × OCH_3_), 69.7 (C-4′), 73.1 (C-2′), 78.4 (C-3′), 80.7 (C-5′), 86.0 (C-1′), 105.7, 107.5, 122.3, 146.4, 149.1, 156.0, 160.5, 161.8, 172.7 (C=O), 179.4 (C=O). HRMS (ESI): *m*/*z* [M − H]^−^ calcd. for C_19_H_21_O_12_S: 473.0759; found: 473.0760.

#### 4.1.5. General Procedure of Methylation of 2-Hydroxy 3-(Tetra-*O*-acetyl-β-d-glucopyranosyl-1-thiomethyl)-1,4-naphthoquinone **28**–**32** to 2-Methoxy Derivatives **43**–**47** Using Diazomethane (Scheme 2)

To solution of 2-hydroxy-3-(tetra-*O*-acetyl-β-d-glucopyranosyl-1-thiomethyl)-1,4-naphthoquinone **28**–**32** (0.50 mmol) in ethylacetate (20 mL) was added dropwise an ethereal solution of diazomethane until TLC (system A) indicated the disappearance of the starting quinone. The reaction mixture was evaporated in vacuo and the residue was crystallized with MeOH, yielding 2-methoxyderivatives **43**–**47** as a red crystals.

3-(Tetra-*O*-acetyl-β-d-glucopyranosyl-1-thiomethyl)-5,8-dihydroxy-2-methoxynaphthalene-1,4- dione **43** (U-634). Red solid; yield 286 mg (96%); *R_f_* = 0.59(B); m.p. 66–67 °C. IR (CHCl_3_): 1755, 1606, 1571, 1456, 1410, 1375, 1293, 1255, 1181, 1141, 1080, 1039 cm^−1^. ^1^H NMR ^1^H (500 MHz, CDCl_3_): δ 1.99 (s, 3H, COCH_3_), 2.02 (s, 6H, 2 × COCH_3_), 2.01 (s, 3H, COCH_3_), 3.65 (ddd, 1H, H-5, *J* = 2.5, 4.5, 10.0 Hz), 3.74 (d, 1H, CH_2_-S, *J* = 13.1 Hz), 3.88 (d, 1H, CH_2_-S, *J* = 13.1 Hz), 3.95 (dd, 1H, H-6′a, *J* = 2.5, 12.1 Hz), 4.14 (dd, 1H, H-6′b, *J* = 4.5, 12.1 Hz), 4.27 (s, 3H, CH_3_O), 4.68 (d, 1H, H-1′, *J* = 10.1 Hz), 5.04 (m, 1H, H-2′), 5.08 (m, 1H, H-4′), 5.21 (m, 1H, H-3′), 7.22 (d, 1H, ArH, *J* = 9.1 Hz), 7.26 (d, 1H, ArH, *J* = 9.1 Hz), 12.30 (s, 1H, α-OH), 12.65 (s, 1H, α-OH).^13^C NMR ^1^H (125 MHz, CDCl_3_): δ 20.54 (COCH_3_), 20.58 (COCH_3_), 20.62 (COCH_3_), 20.65 (COCH_3_), 22.2 (CH_2_S), 61.8 (C-6′) 62.0 (CH_3_O), 68.3 (C-4′), 70.0 (C-2′), 73.9 (C-3′), 75.8 (C-5′), 84.0 (C-1′), 110.8, 111.7, 129.2, 130.5, 133.0, 157.8, 159.0, 169.3 (2 × CH_3_CO), 170.2 (CH_3_CO), 170.4 (CH_3_CO), 183.2 (C=O), 186.8 (C=O). HRMS (ESI): *m*/*z*) [M − H]^−^ calcd. for C_26_H_27_O_14_S:595.1127; found 595.1126.

3-(Tetra-*O*-acetyl-β-d-glucopyranosyl-1-thiomethyl)-5,8-dihydroxy-2-methoxy-6,7-dimethylnaphthalene-1,4-dione **44**. (U-521) Red solid; yield 299 mg (97%); *R_f_* = 0.55 (B); m.p. 105–107 °C. IR (CHCl_3_): 1755, 1598, 1453, 1425, 1397, 1376, 1298, 1243, 1194, 1096, 1042 cm^−1^. ^1^H NMR (500 MHz, CDCl_3_): δ 2.00 (s, 6H, 2 × COCH_3_), 2.01 (s, 3H, COCH_3_), 2.03 (s, 3H, COCH_3_), 2.25 (s, 3H, ArCH_3_), 2.26 (s, 3H, ArCH_3_), 3.66 (ddd, 1H, H-5′, *J* = 2.6, 4.5, 10.1 Hz), 3.77 (d, 1H, CH_2_-S, *J* = 13.0 Hz), 3.96 (d, 1H, CH_2_-S, *J* = 13.0 Hz), 3.97 (dd, 1H, H-6′a, *J* = 2.6, 12.3 Hz), 4.17 (dd, 1H, H-6′b, *J* = 4.5, 12.3 Hz), 4.20 (s, 3H, CH_3_O), 4.68 (d, 1H, H-1′, *J* = 10.2 Hz), 5.02 (m, 1H, H-2′), 5.08 (m, 1H, H-4′), 5.20 (m, 1H, H-3′), 13.11 (s, 1H, α-OH), 13.39 (s, 1H, α-OH).^13^C NMR ^1^H (125 MHz, CDCl_3_): δ 12.3 (ArCH_3_), 12.5 (ArCH_3_), 20.5 (CH_3_CO), 20.6 (2×CH_3_CO), 20.7 (CH_3_CO), 22.4 (CH_2_S), 61.8 (CH_3_O), 61.9 (C-6′), 68.4 (C-4′), 70.1 (C-2′), 74.0(C-3′), 75.8 (C-5′), 83.9 (C-1′), 107.9, 110.0, 131.8, 140.0, 141.5, 156.9, 167.6, 169.0, 169.3 (CH_3_CO), 169.4 (CH_3_CO), 170.2 (CH_3_CO), 170.5 (CH_3_CO), 172.0 (C=O), 176.3 (C=O). HRMS (ESI): *m*/*z* [M − H]^−^ calcd. for C_28_H_31_O_14_S: 623.1440; found 623.1446.

3-(Tetra-*O*-acetyl-β-d-glucopyranosyl-1-thiomethyl)-6,7-dichloro-5,8-dihydroxy-2-methoxynaphthalene-1,4-dione **45** (U-523). Red solid; yield 304 mg (92%); *R_f_* = 0.53 (B); m.p. 126–127 °C. IR (CHCl_3_): 1755, 1608, 1562, 1450, 1404, 1375, 1296, 1250, 1187, 1111, 1040 cm^−1^. ^1^H NMR (500 MHz, CDCl_3_): δ 1.99 (s, 3H, COCH_3_), 2.00 (s, 3H, COCH_3_), 2.01 (s, 3H, COCH_3_), 2.02 (s, 3H, COCH_3_), 3.64 (ddd, 1H, H-5′, *J* = 2.7, 4.5, 10.0 Hz), 3.77 (d, 1H, CH_2_-S, *J* = 13.4 Hz), 3.96 (d, 1H, CH_2_-S, *J* = 13.4 Hz), 3.99 (dd, 1H, H-6a′, *J* = 2.7, 12.5 Hz), 4.14 (dd, 1H, H-6b′, *J* = 4.5, 12.5 Hz), 4.27 (s, 3H, CH_3_O), 4.64 (d, 1H, H-1′, *J* = 10.2 Hz), 5.03 (m, 1H, H-2′), 5.07 (m, 1H, H-4′), 5.20 (m, 1H, H-3′), 12.85 (s, 1H, α-OH), 13.24 (s, 1H, α-OH).^13^C NMR ^1^H (125 MHz, CDCl_3_): δ 20.5 (2 × CH_3_CO), 20.6 (2 × CH_3_CO), 22.0 (CH_2_S), 61.7 (C-6′), 62.3 (CH_3_O), 68.3 (C-4′), 70.0 (C-2′), 73.8 (C-3′), 75.9 (C-5′), 83.8 (C-1′), 108.6, 110.3, 132.5, 134.8, 136.1, 157.4, 159.4, 160.4, 169.3 (CH_3_CO), 169.4 (CH_3_CO), 170.1 (CH_3_CO), 170.4 (CH_3_CO), 176.5 (C=O), 180.5 (C=O). HRMS (ESI): *m*/*z* [M − H]^−^ calcd. for C_26_H_25_Cl_2_O_14_S: 663.0348; found 663.0346.

3-(Tetra-*O*-acetyl-β-d-glucopyranosyl-1-thiomethyl)-5,8-dihydroxy-2,7-dimethoxynaphthalene-1,4-dione **46** (U-645).Red solid; yield 303 mg(97%); *R_f_* = 0.62 (C); m.p. 127–129 °C. IR (CHCl_3_):3083, 2946, 1755, 1606, 1475, 1457, 1436, 1408, 1368, 1302, 1284, 1257, 1180, 1153, 1099, 1039 cm^−^^1^. ^1^H NMR (700 MHz, CDCl_3_):δ1.99 (s, 6H, 2 × COCH_3_), 2.01 (s, 3H, COCH_3_), 2.04 (s, 3H, COCH_3_), 3.66 (ddd, *J* = 9.5, 4.5, 2.5 Hz, 1H, H-5′), 3.81 (d, *J* = 13.1 Hz, 1H, CH_2_-S), 3.95 (s, 3H, OCH_3_), 3.99 (dd, *J* = 12.4, 2.5 Hz, 1H, H-6′a), 4.02 (d, *J* = 13.1 Hz, 1H, CH_2_-S), 4.15 (s, 3H, OCH_3_), 4.19 (dd, *J* = 12.4, 4.5 Hz, 1H, H-6′b), 4.69 (d, *J* = 10.2 Hz, 1H, H-1′), 5.03 (dd, *J* = 10.2, 9.5 Hz, 1H, H-2′), 5.09 (t, *J* = 9.5 Hz, 1H, H-4′), 5.20 (t, *J* = 9.5 Hz, 1H, H-3′), 6.29 (s, 1H, H-3), 12.73 (s, 1H, α-OH), 13.33 (s, 1H, α-OH).^13^C NMR (175 MHz, CDCl_3_):δ20.6 (2 × COCH_3_), 20.7 (2 × COCH_3_), 22.5 (CH_2_-S), 56.8 (OCH_3_), 61.7 (OCH_3_), 61.9 (C-6′), 68.3 (C-4′), 70.1 (C-2′), 74.0 (C-3′), 75.8 (C-5′), 83.9 (C-1′), 105.9, 109.4, 111.6, 132.8, 155.0, 159.9, 162.2, 165.2, 169.3 (2 × COCH_3_), 170.2 (COCH_3_), 170.5 (COCH_3_), 173.6 (C=O), 179.4 (C=O). HRMS (ESI): *m*/*z* [M − H]^−^ calcd. for C_17_H_29_O_15_S: 625.1233; found: 625.1226.

3-(Tetra-*O*-acetyl-β-d-glucopyranosyl-1-thiomethyl)-5,8-dihydroxy-2,6,7-trimethoxynaphthalene-1,4-dione **47** (U-638). Red solid; yield 311 mg (95%); *R_f_* = 0.65 (C); m.p. 135–137 °C. IR (CHCl_3_):3083, 2950, 1755, 1602, 1458, 1408, 1376, 1287, 1257, 1151, 1045 cm^−^^1^. ^1^H NMR (500 MHz, CDCl_3_):δ1.99 (s, 6H, 2 × COCH_3_), 2.01 (s, 3H, COCH_3_), 2.04 (s, 3H, COCH_3_), 3.66 (ddd, 1H, H-5′, *J* = 9.5, 4.5, 2.5 Hz), 3.81 (d, 1H, CH_2_-S, *J* = 13.1 Hz), 4.01 (dd, 1H, H-6′a, *J* = 12.4, 2.5 Hz), 4.03 (d, 1H, CH_2_-S, *J* = 13.1 Hz), 4.10 (s, 3H, OCH_3_), 4.14 (s, 3H, OCH_3_), 4.15 (s, 3H, OCH_3_), 4.18 (dd, 1H, H-6′b, *J* = 12.4, 4.5 Hz), 4.66 (d, 1H, H-1′,*J* = 10.2 Hz), 5.02 (dd, 1H, H-2′, *J* = 10.2, 9.5 Hz), 5.09 (t, 1H, H-4′, *J* = 9.5 Hz), 5.19 (t, 1H, H-3′, *J* = 9.5 Hz), 12.91 (s, 1H, α-OH), 13.12 (s, 1H, α-OH).^13^C NMR (125 MHz, CDCl_3_):δ 20.5 (2 × COCH_3_), 20.6 (2 × COCH_3_), 22.4 (CH_2_-S), 61.6 (OCH_3_), 61.7 (2×OCH_3_), 61.9 (C-6′), 68.4 (C-4′), 70.0 (C-2′), 74.0 (C-3′), 75.8 (C-5′), 83.8 (C-1′), 106.3, 109.8, 130.6, 147.5, 148.6, 155.7, 157.9, 163.0, 169.3 (2 × COCH_3_), 170.2 (COCH_3_), 170.5 (COCH_3_), 178.2 (C=O), 179.6 (C=O). HRMS (ESI): *m*/*z* [M − H]^−^ calcd. for C_28_H_31_O_16_S: 655.1338; found: 655.1332.

Deacetylation of 3-(tetra-*O*-acetyl-β-d-glucopyranosyl-1-thiomethyl)-5,8-dihydroxy-2-methoxynaphthalene-1,4-diones **43**–**47** was done according Section 4.1.4. General Procedure and led to thiomethyl glucosides **48**–**52** (Scheme 2)

3-(β-d-Glucopyranosyl-1-thiomethyl)-5,8-dihydroxy-3-methoxynaphthalene-1,4-dione **48** (U-636). Red solid; yield 62 mg (72%); *R_f_* = 0.37 (D); m.p. 103–105 °C. IR (KBr): 3417, 2922, 1608, 1455, 1409, 1385, 1283, 1181, 1110, 1077, 1029, 789 cm^−1^. ^1^H NMR ^1^H (500 MHz, DMSO-*d*_6_): δ 2.96 (m, 1H, H-2′), 3.05 (m, 1H, H-5′), 3.08 (m, 1H, H-4′), 3.11 (m, 1H, H-3′), 3.39 (dd, 1H, H-6′a, *J* = 5.4, 12.0 Hz), 3.55 (dd, 1H, H-6′b, *J* = 1.7, 12.0 Hz), 3.60 (d, 1H, CH_2_-S, *J* = 13.2 Hz), 3.83 (d, 1H, CH_2_-S, *J* = 13.2 Hz), 4.15 (s, 3H, OCH_3_), 4.32 (d, 1H, H-1′, *J* = 9.9 Hz), 7.35 (d, 1H, ArH, *J* = 9.4 Hz), 7.38 (d, 1H, ArH, *J* = 9.4 Hz), 12.00 (s, 1H, α-OH), 12.50 (s, 1H, α-OH). ^13^C NMR ^1^H (125 MHz, DMSO-*d*_6_): δ 21.0 (CH_2_-S), 61.0 (C-6′), 61.8 (OCH_3_), 69.9 (C-4′), 73.0 (C-2′), 78.4(C-3′), 81.0 (C-5′), 85.4 (C-1′), 111.0, 111.8, 128.8, 129.7, 133.0, 156.2, 156.9, 157.7, 183.4 (C=O), 187.3 (C=O). HRMS (ESI): *m*/*z* [M − H]^−^ calcd. for C_18_H_19_O_10_S:427.0704; found 427.0700.

3-(β-d-Glucopyranosyl-1-thiomethyl)-5,8-dihydroxy-2-methoxy-6,7-dimethylnaphthalene-1,4-dione **49** (U-522). Red solid; yield 74 mg (81%); *R_f_* = 0.53 (D); m.p. 209–211 °C. IR (KBr): 3391, 2923, 1605, 1582, 1454, 1421, 1384, 1334, 1300, 1281, 1191, 1137, 1086, 1047, 809 cm^−1^. ^1^H NMR ^1^H (700 MHz, DMSO-*d*_6_): δ 2.20 (s, 3H, ArCH_3_), 2.21 (s, 3H, ArCH_3_), 2.96 (m, 1H, H-2′), 3.06 (m, 1H, H-5′), 3.09 (m, 2H, H-2′, H-4′), 3.41 (m, 1H, H-6′a), 3.57 (m, 1H, H-6′b), 3.64 (d, 1H, CH_2_-S, *J* = 13.2 Hz), 3.88 (d, 1H, CH_2_-S, *J* = 13.2 Hz), 4.11 (c, 3H, CH_3_O), 4.33 (d, 1H, H-1′, *J* = 10.0 Hz), 4.41 (t, 1H, CH_2_OH, *J* = 5.9 Hz), 4.86 (d, 1H, CH-OH, *J* = 4.5 Hz), 4.97 (d, 1H, CH-OH, *J* = 3.9 Hz), 5.02 (d, 1H, CH-OH, *J* = 6.1 Hz), 12.91 (s, 1H, α-OH), 13.29 (s, 1H, α-OH). ^13^C NMR ^1^H (175 MHz, DMSO-*d*_6_): δ 12.1 (ArCH_3_), 12.3(ArCH_3_), 21.1 (CH_2_-S), 61.0 (C-6′), 61.8 (MeO), 69.9 (C-4′), 73.0 (C-2′), 78.4 (C-3′), 81.0 (C-5′), 85.3 (C-1′), 107.9, 109.5, 132.6, 138.8, 140.1, 156.7, 164.0, 165.0, 174.5 (C=O), 178.8 (C=O). HRMS (ESI): *m*/*z* [M − H]^−^ calcd. for C_20_H_23_O_10_S: 455.1017; found 455.1018.

6,7-Dichloro-3-(β-d-glucopyranosyl-1-thiomethyl)-5,8-dihydroxy-2-methoxynaphthalene-1,4-dione **50** (U-625) Red solid; yield 66 mg (67%); *R_f_* = 0.48 (D); m.p. 180–182 °C. IR (KBr): 3417, 2895, 1611, 1587, 1452, 1402, 1333, 1279, 1227, 1180, 1115, 1043, 1017, 906, 855 cm^−1^. ^1^H NMR ^1^H (500 MHz, DMSO-*d*_6_): δ 2.96 (dd, 1H, H-2′, *J* = 8,3, 9.7 Hz), 3.05 (m, 2H, H-4′, H-5′), 3.11 (m, 1H, H-3′), 3.37 (dd, 1H, H-6′a, *J* = 4.4, 12.1 Hz), 3.57 (d,1H, H-6′b, *J* = 12.1 Hz), 3.66 (d, 1H, CH_2_-S, *J* = 13.1), 3.86 (d, 1H, CH_2_-S, *J* = 13.1), 4.16 (s, 3H, OCH_3_), 4.33 (d, 1H, H-1′, *J* = 9.7 Hz), 12.43 (s, 1H, α-OH), 13.01 (s, 1H, α-OH). ^13^C NMR ^1^H (125 MHz, DMSO-*d*_6_): δ 21.2 (CH_2_-S), 61.1 (C-6′), 62.0 (OCH_3_), 70.0 (C-4′), 73.0 (C-2′), 78.3 (C-3′), 81.0 (C-5′), 85.4 (C-1′), 109.8, 111.3, 132.4, 132.5, 133.1, 156.9, 157.1, 157.5, 177.2 (C=O), 181.6 (C =O). HRMS (ESI): *m*/*z* [M − H]^−^ calcd. for C_18_H_17_Cl_2_O_10_S: 494.9925; found 494.9922.

3-(β-d-Glucopyranosyl-1-thiomethyl)-5,8-dihydroxy-2,7-dimethoxynaphthalene-1,4-dione **51** (U-646). Red solid; yield 62 mg (67%); *R_f_* = 0.44 (E); m.p. 141–143 °C. IR (KBr):3402, 2936, 2361, 1603, 1451, 1430, 1403, 1385, 1291, 1223, 1192, 1180, 1142, 1100, 1074, 1032, 955 cm^−^^1^. ^1^H NMR (500 MHz, DMSO-*d*_6_):δ2.96 (m, 1H, H-2′), 3.06-3.12 (m, 3H, H-3′,H-4′, H-5′), 3.43 (dd, *J* = 11.8, 4.8 Hz, 1H, H-6′a), 3.60 (dd, *J* = 11.8, 2.0 Hz, 1H, H-6′b), 3.69 (d, *J* = 13.0 Hz, 1H, CH_2_-S), 3.92 (s, 3H, OCH_3_), 3.94 (d, 1H, CH_2_-S, *J* = 13.0 Hz), 4.03 (s, 3H, OCH_3_), 4.34 (d, 1H, H-1′, *J* = 9.8 Hz), 4.42 (br s, 1H, OH′), 4.87 (br s, 1H, OH′), 4.98 (br s, 1H, OH′), 5.03 (br s, 1H, OH′), 6.58 (s, 1H, H-6), 12.48 (s, 1H, α-OH), 13.32 (s, 1H, α-OH). ^13^C NMR (125 MHz, DMSO-*d*_6_):δ 21.2 (CH_2_-S), 57.1 (OCH_3_), 61.1 (C-6′), 61.5 (OCH_3_), 70.0 (C-4′), 73.1 (C-2′), 78.4 (C-3′), 81.1 (C-5′), 85.3 (C-1′), 105.7, 109.3, 111.4, 133.3, 154.6, 159.7, 163.3, 166.5, 170.9 (C=O), 177.7 (C=O). HRMS (ESI): *m*/*z* [M − H]^−^ calcd. for C_19_H_21_O_11_S: 457.0810; found: 457.0812.

3-(β-d-Glucopyranosyl-1-thiomethyl)-5,8-dihydroxy-2,6,7-trimethoxynaphthalene-1,4-dione **52** (U-641). Red solid; 74 mg (76%); *R_f_* = 0.51 (E); m.p. 155–157 °C. IR (KBr):3415, 2950, 1601, 1455, 1403, 1385, 1275, 1211, 1180, 1142, 1102, 1068, 1049, 1023, 943, 876 cm^−^^1^. ^1^H NMR (700 MHz, DMSO-*d*_6_):δ2.96 (m, 1H, H-2′), 3.06-3.12 (m, 3H, H-3′,H-4′, H-5′), 3.44 (dd, 1H, H-6′a, *J* = 11.8, 2.0 Hz), 3.62 (dd, 1H, H-6′b, *J* = 11.8, 2.0 Hz), 3.68 (d, *J* = 13.0 Hz, 1H, CH_2_-S), 3.95 (d, *J* = 13.0 Hz, 1H, CH_2_-S), 3.99 (s, 3H, OCH_3_), 4.02 (s, 3H, OCH_3_), 4.03 (s, 3H, OCH_3_), 4.32 (d, 1H, H-1′, *J* = 9.8 Hz), 4.46 (br s, 1H, OH′), 4.88 (br s, 1H, OH′), 4.98 (br s, 1H, OH′), 5.04 (br s, 1H, OH′), 12.71 (br s, 1H, α-OH), 12.97 (s, 1H, α-OH). ^13^C NMR (175 MHz, DMSO-*d*_6_):δ 21.2 (CH_2_-S), 61.2 (C-6′), 61.4 (2×OCH_3_), 61.6 (OCH_3_), 70.1 (C-4′), 73.1 (C-2′), 78.5 (C-3′), 81.1 (C-5′), 85.3 (C-1′), 106.5, 109.6, 131.6, 147.6, 148.4, 155.1, 158.4, 163.5, 176.7 (C=O), 177.8 (C=O). HRMS (ESI): *m*/*z* [M − H]^−^ calcd. for C_20_H_23_O_12_S: 487.0916; found: 487.0918.

#### 4.1.6. General Procedure for Synthesis of Spinochrome D Acetylated Thiomethylglycosides **56**–**59** by Acid-Catalytic Condensation of Spinochrome D **8** with per-O-acetyl-1-thiomercaptho Derivatives of d-Glucopyranose **27**, d-Galactopyranose **53**, d-Mannopyranose **54**, d-Xylopyranose **55** and Paraformaldehyde in Dioxane (Figure 4)

Spinochrome D **8** (119 mg, 0.50 mmol) was dissolved in a warm mixture of 1,4-dioxane (20 mL), water (3 mL), and acetic acid (0.2 mL), to which powdered paraformaldehyde (60 mg, 2.00 mmol) and corresponding per-*O*-acetylated 1-mercaptoshugar **27**, **53**–**55** (0.65 mmol) were added. The reaction mixture was gently refluxed with mixing for 3h, concentrated in vacuo, and the resulting solid was purified by preparative TLC (silica gel, system C, two developments), yielding a red-colored fraction of thiomethylated product **56**–**59** with *R_f_* = 0.55–0.57 (Figure 4).

3-(Tetra-*O*-acetyl-β-d-glucopyranosyl-1-thiomethyl)-2,5,6,7,8-pentahydroxynaphthalene-1,4-dione **56** (U-629). Red solid; 212 mg (69.0%); *R_f_* = 0.55 (C); m.p. 127–129 °C. IR (CHCl_3_): 3524, 3421, 3053, 3007, 1748, 1715, 1601, 1458, 1428, 1367, 1294, 1247, 1229, 1221, 1200, 1187, 1090, 1044 cm^−^^1^. ^1^H NMR (700 MHz, CDCl_3_):δ 2.00 (s, 6H, 2 × COCH_3_), 2.02 (s, 3H, COCH_3_), 2.06 (s, 3H, COCH_3_), 3.69 (m, 1H, H-5′), 3.82 (d, 1H, *J* = 13.6 Hz, CH_2_S), 4.04 (dd, 1H, H-6′a,*J* = 12.3, 2.9 Hz), 4.06 (d, 1H, CH_2_S, *J* = 13.6 Hz), 4.22 (dd, 1H, *J* = 12.3, 4.6 Hz), 4.68 (d, 1H, *J* = 10.2 Hz, H-1′), 5.05 (t, 1H, *J* = 9.7 Hz, H-2′), 5.09 (t, 1H, H-4′, *J* = 9.7 Hz), 5.22 (t, 1H, H-3′, *J* = 9.2 Hz), 6.55 (s, 1H, β-OH), 6.91 (s, 1H, β-OH), 7.19 (s, 1H, β-OH), 12.02 (brs, 1H, α-OH), 12.36 (s, 1H, α-OH). ^13^C NMR (176 MHz, CDCl_3_): δ20.6 (3 × COCH_3_), 20.7 (COCH_3_), 21.8 (CH_2_-S), 62.3 (C-6′), 68.7 (C-4′), 70.1 (C-2′), 73.8 (C-3′), 75.7 (C-5′), 83.8 (C-1′), 102.4, 107.6, 121.1, 137.2, 139.2, 150.1, 152.4, 161.4, 169.4 (2 × COCH_3_), 170.2 (COCH_3_), 170.8 (COCH_3_), 177.8 (C=O), 179.5 (C=O). HRMS (ESI): *m*/*z* [M − H]^−^ calcd. for C_25_H_25_O_16_S 613.0869; found 613.0865.

3-(Tetra-*O*-acetyl-β-d-galactopyranosyl-1-thiomethyl)-2,5,6,7,8-pentahydroxynaphthalene-1,4-dione **57** (U-631). Red solid; 204 mg (66.5%); *R_f_* = 0.55 (C); m.p. 135–137 °C. IR (CHCl_3_): 3523, 3432, 3054, 3007, 1750, 1687, 1590, 1465, 1429, 1371, 1294, 1248, 1188, 1150, 1084, 1055 cm^−^^1^. ^1^H NMR (700 MHz, DMSO-*d*_6_):δ 1.90 (s, 3H, COCH_3_), 1.95 (s, 3H, COCH_3_), 1.97 (s, 3H, COCH_3_), 2.11 (s, 3H, COCH_3_), 3.72 (d, 1H, CH_2_S, *J* = 12.9 Hz), 3.88 (d, 1H, CH_2_S, *J* = 12.9 Hz), 3.90 (dd, 1H, H-6′a, *J* = 11.1, 6.5 Hz), 3.97 (dd, 1H, H-6′b, *J* = 11.1, 6.3 Hz), 4.14 (m, 1H, H-5′), 4.91 (d, 1H, H-1′, *J* = 10.2 Hz), 4.94 (t, 1H, H-2′, *J* = 9.8 Hz), 5.20 (dd, 1H, H-3′, *J* = 9.5, 3.6 Hz), 5.29 (dd, 1H, H-4′, *J* = 3.6, 0.9 Hz), 10.31 (s, 3H, 3 × β-OH), 12.60 (brs, 1H, α-OH), 13.29 (s, 1H, α-OH). ^13^C NMR (176 MHz, DMSO-*d*_6_): δ20.3 (COCH_3_), 20.4 (2 × COCH_3_), 20.5 (COCH_3_), 21.8 (CH_2_-S), 61.1 (C-6′), 67.5 (C-4′), 67.6 (C-2′), 71.1 (C-3′), 73.4 (C-5′), 83.0 (C-1′), 102.1, 107.0, 119.7, 139.8, 142.4, 155.3, 157.7, 166.1, 169.3 (COCH_3_), 169.4 (COCH_3_), 169.8 (COCH_3_), 170.0 (COCH_3_), 172.1 (C=O), 173.8 (C=O). HRMS (ESI): *m*/*z* [M − H]^−^ calcd. for C_25_H_25_O_16_S 613.0869; found 613.0858.

3-(Tetra-*O*-acetyl-β-d-mannopyranosyl-1-thiomethyl)-2,5,6,7,8-pentahydroxynaphthalene-1,4-dione **58** (U-630). Red solid; 200 mg (65.0%); *R_f_* = 0.55 (C); m.p. 160–162 °C. IR (CHCl_3_): 3524, 3446, 3054, 3007, 1749, 1686, 1637, 1600, 1541, 1508, 1458, 1430, 1369, 1293, 1247, 1187, 1104, 1052 cm^−^^1^. ^1^H NMR (500 MHz, CDCl_3_): δ 1.96 (s, 3H, COCH_3_), 2.04 (s, 3H, COCH_3_), 2.09 (s, 3H, COCH_3_), 2.16 (s, 3H, COCH_3_), 3.68 (m, 1H, H-5′), 3.83 (d, 1H, CH_2_S, *J* = 13.7 Hz), 4.07 (d, 1H, CH_2_S, *J* = 13.7 Hz), 4.10 (dd, 1H, H-6′a, *J* = 12.2, 2.2 Hz), 4.30 (dd, 1H, H-6′b, *J* = 12.2, 5.2 Hz), 4.90 (d, 1H, H-1′, *J* = 1.0 Hz), 5.05 (d.d, 1H, H-3′, *J* = 10.0, 3.5 Hz), 5.27 (t, 1H, H-4′, *J* = 10.0 Hz), 5.46 (dd, 1H, H-2′, *J* = 3.6, 1.0 Hz), 6.50 (s, 1H, β-OH), 6.87 (s, 1H, β-OH), 7.12 (s, 1H, β-OH), 12.01 (brs, 1H, α-OH), 12.33 (s, 1H, α-OH). ^13^C NMR (125 MHz, CDCl_3_): δ20.5 (2 × COCH_3_), 20.7 (COCH_3_), 20.8 (COCH_3_), 22.4 (CH_2_-S), 63.0 (C-6′), 66.4 (C-4′), 70.3 (C-2′), 71.9 (C-3′), 76.3 (C-5′), 82.4 (C-1′), 102.6, 107.7, 121.1, 137.1, 139.1, 149.8, 152.4, 161.0, 169.6 (COCH_3_), 170.1 (COCH_3_), 170.2 (COCH_3_), 170.9 (COCH_3_), 178.0 (C=O), 179.7 (C=O). HRMS (ESI): *m*/*z* [M − H]^−^ calcd. for C_25_H_25_O_16_S 613.0869; found 613.0866.

3-(Tri-*O*-acetyl-β-d-xylopyranosyl-1-thiomethyl)-2,5,6,7,8-pentahydroxynaphthalene-1,4-dione **59** (U-628).Red solid; 171 mg (63.0%); *R_f_* = 0.57 (C); m.p. 235–237 °C. IR (CHCl_3_): 3467, 3342, 3046, 3009, 1732, 1592, 1422, 1376, 1330, 1287, 1267, 1241, 1220, 1208, 1200 cm^−^^1^. ^1^H NMR (500 MHz, DMSO-*d*_6_): δ 1.94 (s, 3H, COCH_3_), 1.97 (s, 3H, COCH_3_), 1.99 (s, 3H, COCH_3_), 3.45 (d.d, 1H, H-5′a, *J* = 11.4, 9.8 Hz), 3.71 (d, 1H, CH_2_S, *J* = 12.7 Hz), 3.85 (d, 1H, CH_2_S, *J* = 12.7 Hz), 3.98 (dd, 1H, H-5′b, *J* = 11.4, 5.3 Hz), 4.81 (t, 1H, H-2′, *J* = 9.0), 4.84 (m, 1H, H-4′), 4.88 (d, 1H, H-1′, *J* = 9.2 Hz), 5.16 (t, 1H, H-3′, *J* = 8.8 Hz), 10.46 (brs, 3H, 3 × β-OH), 12.48 (brs, 1H, α-OH),13.35 (s, 1H, α-OH). ^13^C NMR (125 MHz, DMSO-*d*_6_): δ20.4 (2 × COCH_3_), 20.5 (COCH_3_), 21.7 (CH_2_-S), 64.9 (C-5′), 68.5 (C-4′), 69.9 (C-2′), 72.2 (C-3′), 83.1(C-1′), 102.2, 107.0, 119.5, 139.7, 142.4, 155.5, 158.5, 166.7, 169.1 (COCH_3_), 169.5 (2 × COCH_3_), 171.3 (C=O), 173.0 (C=O). HRMS (ESI): *m*/*z* [M − H]^−^ calcd. for C_22_H_21_O_1__4_S 541.0657; found 541.0651.

#### 4.1.7. General Procedure for Base-Catalytic Deacetylation of Acetylated Thiomethylglycosides Spinochrome D **56**–**59** in MeONa/Methanol Solution (Figure 4)

Acetylated thiomethylglycosides of spinochrome D **56**–**59** (0.25 mmol) were suspended in dry MeOH (10 mL) and (2.0 mL, 1.0 mM) of 0.5 N MeONa/MeOH solution was added under argon atmosphere. The reaction mixture was kept at room temperature for 1 h and then acidified with 2N HCl to give a clear red solution. The reaction mixture was concentrated in vacuoand the resulting solid was purified by preparative TLC (system E), yielding a red-colored fraction of desacetylated glycosides **60**–**63** with *R_f_* = 0.30–0.40 (Figure 4).

3-(β-d-Glucopyranosyl-1-thiomethyl)-2,5,6,7,8-pentahydroxynaphthalene-1,4-dione **60** (U-649). Red solid; 81 mg (72.6%); *R_f_* = 0.33 (E); m.p. 177–180 °C. IR (KBr): 3402, 2923, 1588, 1468, 1427, 1385, 1285, 1097, 1046, 983, 876, 786, 766, 719 cm^−^^1^. ^1^H NMR (700 MHz, DMSO-*d*_6_): δ 2.96 (dd, 1H, H-2′, *J* = 9.7, 8.5 Hz), 3.05 (m, 1H, H-5′), 3.10 (t, 1H, H-3′, *J* = 8.5 Hz), 3.13 (m, 1H, H-4′), 3.45 (dd, 1H, H-6′a, *J* = 11.8, 5.0 Hz), 3.57 (dd, 1H, H-6′b*,J* = 11.9, 2.0 Hz), 3.68 (d, 1H, CH_2_S, *J* = 13.0 Hz), 3.86 (d, 1H, CH_2_S, *J* = 13.0 Hz), 4.39 (d, 1H, H-1′, *J* = 9.8 Hz), 4.90 (br s, 4H, carbohydr. hydroxyls), 10.39 (br s, 3H, 3 × β-OH), 12.50 (br s, 1H, α-OH), 13.29 (br s, 1H, α-OH). ^13^C NMR (176 MHz, DMSO-*d*_6_): δ21.2 (CH_2_-S), 60.8 (C-6′), 69.7 (C-4′), 73.2 (C-2′), 78.5 (C-3′), 80.7 (C-5′), 85.9 (C-1′), 102.2, 106.8, 121.2, 139.9, 142.4, 154.9, 158.0, 166.5, 172.0 (C=O), 173.6 (C=O). HRMS (ESI): *m*/*z* [M − H]^−^ calcd. for C_17_H_17_O_1__2_S 445.0450; found 445.0451.

3-(β-d-Galactopyranosyl-1-thiomethyl)-2,5,6,7,8-pentahydroxynaphthalene-1,4-dione **61** (U-650). Red solid; 75 mg (67.3%); *R_f_* = 0.30 (E); m.p. 186–189 °C. IR (KBr): 3345, 2925, 1587, 1469, 1425, 1385, 1285, 1140, 1083, 1052, 980, 865, 768 cm^−^^1^. ^1^H NMR (500 MHz, DMSO-*d*_6_): δ 3.23 (dd, 1H, H-3′, *J* = 9.0, 3.2 Hz), 3.27 (t, 1H, H-2′, *J* = 9.2 Hz), 3.31 (m, 1H, H-5′), 3.38 (dd, 1H, H-6′a, *J* = 10.5, 5.8 Hz), 3.48 (dd, 1H, H-6′b, *J* = 10.5, 7.0 Hz), 3.70 (m, 1H, H-4′), 3.70 (d, 1H, CH_2_S, *J* = 12.8 Hz), 3.83 (d, 1H, CH_2_S, *J* = 12.8 Hz), 4.37 (d, 1H, H-1′, *J* = 9.5 Hz), 4.71 (br s, 4H, carbohydr. hydroxyls), 10.37 (br.s, 3H, 3 × β-OH), 12.34 (br s, 1H, α-OH), 13.30 (br s, 1H, α-OH). ^13^C NMR (125 MHz, DMSO-*d*_6_): δ21.3 (CH_2_-S), 60.0 (C-6′), 68.1 (C-4′), 70.2 (C-2′), 74.9 (C-3′), 78.9 (C-5′), 86.5 (C-1′), 102.2, 106.8, 121.1, 128.4, 139.8, 142.4, 154.9, 157.8, 166.3, 172.2 (C=O), 173.9 (C=O). HRMS (ESI): *m*/*z* [M − H]^−^ calcd. for C_17_H_17_O_1__2_S 445.0450, found 445.0446.

3-(β-d-Mannopyranosyl-1-thiomethyl)-2,5,6,7,8-pentahydroxynaphthalene-1,4-dione **62** (U-648). Red solid; 77 mg (69.0%); *R_f_* = 0.30 (E); m.p. 185–187 °C. IR (KBr): 3394, 2936, 1704, 1591, 1463, 1426, 1385, 1293, 1054, 985, 881, 772 cm^−^^1^. ^1^H NMR (500 MHz, DMSO-*d*_6_): δ 3.00 (m, 1H, H-5′), 3.26 (dd, 1H, H-3′, *J* = 9.3, 3.4 Hz), 3.36 (t, 1H, H-4′, *J* = 9.3 Hz), 3.44 (dd, 1H, H-6′a, *J* = 11.5, 5.1 Hz), 3.51 (dd, 1H, H-6′b, *J* = 11.5, 2.4 Hz), 3.62 (m, 1H, H-2′), 3.76 (s, 2H, CH_2_S), 4.15 (br s, 4H, carbohydr. hydroxyls), 4.75 (d, 1H, H-1′, *J* = 1.2 Hz), 10.14 (br s, 1H, β-OH), 10.48 (br s, 1H, β-OH), 11.19 (br s, 1H, β-OH), 12.72 (br s, 1H, α-OH), 13.22 (br s, 1H, α-OH). ^13^C NMR (DMSO-*d*_6_, 125 MHz): δ22.5 (CH_2_-S), 61.1 (C-6′), 66.6 (C-4′), 72.2 (C-2′), 74.7 (C-3′), 81.2 (C-5′), 85.2 (C-1′), 102.3, 106.8, 121.5, 140.0, 142.3, 154.3, 157.7, 166.4, 172.3 (C=O), 173.8 (C=O). HRMS (ESI): *m*/*z* [M − H]^−^ calcd. for C_17_H_17_O_1__2_S 445.0450; found 445.0446.

3-(β-d-Xylopyranosyl-1-thiomethyl)-2,5,6,7,8-pentahydroxynaphthalene-1,4-dione **63** (U-647). Red solid; 62 mg (59.6%); *R_f_* = 0.40 (E); m.p. 177–179 °C. IR (KBr): 3380, 2921, 1588, 1464, 1426, 1286, 1157, 1136, 1092, 1041, 980, 926, 769, 717, 630 cm^−^^1^. ^1^H NMR (500 MHz, DMSO-*d*_6_): δ 2.96 (dd, 1H, H-2′, *J* = 9.4, 8.3 Hz), 3.02 (dd, 1H, H-5′a, *J* = 11.3, 10.2 Hz), 3.07 (t, 1H, H-3′, *J* = 8.3 Hz), 3.29 (m, 1H, H-4′), 3.65 (d, 1H, CH_2_S, *J* = 13.0 Hz), 3.74 (dd, 1H, H-5′b, *J* = 11.3, 5.3 Hz), 3.83 (d, 1H, CH_2_S, *J* = 13.0 Hz), 4.35 (d, 1H, H-1′,*J* = 9.4 Hz), 4.91 (br s, 3H, carbohydr. hydroxyls), 10.15 (br s, 1H, β-OH), 10.49 (br s, 1H, β-OH), 11.20 (br s, 1H, β-OH), 12.73 (br s, 1H, α-OH), 13.23 (br s, 1H, α-OH). ^13^C NMR (176 MHz, DMSO-*d*_6_): δ21.0 (CH_2_-S), 69.2 (C-5′), 69.5 (C-4′), 73.0 (C-2′), 78.0 (C-3′), 86.3 (C-1′), 102.2, 106.9, 121.2, 140.0, 142.3, 154.4, 157.4, 166.2, 172.5 (C=O), 174.1 (C=O). HRMS (ESI): *m*/*z* [M − H]^−^ calcd. for C_16_H_5_O_1__1_S 415.0341, found 415.0343.

### 4.2. Cell Culture

The cells of the mouse Neuro-2a neuroblastoma (ATCC® CCL-131™; American Type Culture Collection, Manassas, USA) were cultured in DMEM medium containing 10% fetal bovine serum (Biolot, St. Petersburg, Russia) and 1% penicillin/streptomycin (Biolot, St. Petersburg, Russia). The cells were placed in 96-well plates in a concentration of 3 × 10^4^ cells per well and incubated at 37 °C in a humidified atmosphere containing 5% (*v*/*v*) CO_2_.

### 4.3. Cytotoxic Activity Assay

Neuro-2a cells (3 × 10^4^ cells/well) were incubated with different concentrations of 5,8-dihydroxy-1,4-naphthoquinone derivatives in a CO_2_-incubator for 24 h at 37 °C. After incubation, the medium with tested substances was replaced with 100 μL of pure medium. Cell viability was determined using the MTT (3-(4,5-dimethylthiazol-2-yl)-2,5-diphenyltetrazolium bromide) method, according to the manufacturer’s instructions (Sigma-Aldrich, St. Louis, USA). For this purpose, 10 μL of MTT stock solution (5 mg/mL) was added to each well and the microplate was incubated for 4 h at 37 °C. After that, 100 μL of SDS-HCl (1 g SDS/10 ml dH_2_O/17 μL 6 N HCl) was added to each well, followed by incubation for 4–18 h. The absorbance of the converted dye formazan was measured using a Multiskan FC microplate photometer (Thermo Scientific, Waltham, USA) at a wavelength of 570 nm. The results were presented as percent of control data, and the concentration required for 50% inhibition of cell viability (EC_50_) was calculatedusing SigmaPlot 3.02 (Jandel Scientific, San Rafael, CA, USA). All data were obtained in three independent replicates and expressed as mean ± SEM.

Stocks of substances were prepared in DMSO at a concentration of 10 mM. All studied 1,4-NQ derivatives were tested in a concentration range from 0.4 μM to 100 μM with two-fold dilution. All tested compounds were added in a volume of 20 μL dissolved in PBS to 180 μL of cell culture with the cells in each of the wells (final DMSO concentration < 1%). Triterpene glycoside cladoloside C isolated from the sea cucumber *Cladolabes schmeltzii* [60] was used as a reference cytotoxic compound.

### 4.4. Computer Modeling and Quantitative Structure-Activity Relationship (QSAR) 

In the present study, a dataset of 50 1,4-NQs derivatives (Table 1) was used for 3D-structure modeling and optimization with Amber 10:EHT force field using the Build module of the MOE 2019.01 program [61]. The dataset was used for descriptor calculations with the QuaSAR module of MOE 2019.01. The EC_50_ values were converted into corresponding pEC_50_ values (−logEC_50_) to be included in the database. The pEC50 values determined in this work for 22 compounds with 4 < pEC_50_ < 6 were added to the database with the structures of the studied compounds. The dataset of the 22 1,4-NQs derivatives with 4 < pEC_50_ < 6 was divided into training (18) and test (4) sets (Table 3), which were used for the generation of a QSAR model and its validation, respectively. The MOE Pharmacophore editor for pharmacophore modeling was used, which consisted of standard pharmacophoric features including hydrogen bond acceptor (Acc), hydrogen bond donor (Don), hydrophobic (Hyd), and aromatic ring (Aro). The energy-optimized molecules with high toxicity U-634 and nontoxic U-633 were used for the development of the pharmacophore models.

## 5. Conclusions

Based on 6,7-substituted 2,5,8-trihydroxy-1,4-naphtoquinones (1,4-NQs) derived from sea urchins, five new acetyl-*O*-glucosides of NQ were prepared. A new method of conjugation of per-*O*-acetylated 1-mercaptosaccharides with 2-hydroxy-1,4-NQs through a methylene spacer was developed. Methylation of the 2-hydroxy group of the quinone core of acetylthiomethylglycosides by diazamethane and deacetylation of sugar moiety led to the synthesis 28 new thiomethylglycosides of 2-hydroxy- and 2-methoxy-1,4-NQs. The cytotoxic activity of starting 1,4-NQs (13 compounds) and their *O*- and *S*-glycoside derivatives (37 compounds) was determined by the MTT method against Neuro-2a mouse neuroblastoma cells. A computer model of the effect on cancer cells of the chemical structure–activity relationship (QSAR) of 5,8-dihydroxy-1,4-NQ derivatives was constructed. The structural elements of the naphthoquinone molecules (pharmacophores) which determined the high cytotoxic activity were revealed. These results can be taken into account during the directed modification of the quinone structure to obtain new, highly selective compounds which are toxic to tumor cells. QSAR analysis can be used to predict cytotoxic activity and targeted modification of the quinone structure in the preparation of new neuroprotectors for the treatment of brain tumors.

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
