# Peer review of "Synthesis, Cytotoxic Activity Evaluation and Quantitative Structure-ActivityAnalysis of Substituted 5,8-Dihydroxy-1,4-naphthoquinones and Their O- and S-Glycoside Derivatives Tested against Neuro-2a Cancer Cells"

_marinedrugs, 2020, doi:10.3390/md18120602_

Round 1

Reviewer 1 Report

The submitted manuscript describes the synthesis of a number of novel glycosidic 5,8-Dihydroxy-1,4‐Naphthoquinones and their biological testing on a single cancer cell line. There is also a significant study of the molecules theoretically, in an attempt to determine an overall SAR.

The methods of synthesis are well reported and so is the experimental data. The is not much justification for the cell line chosen and should be addressed. Results could be compared to previous work on other cells lines. 

The SAR study concludes not with positional information about groups but mainly concluding that lipophilic properties are the main determinant of activity. I would the authors to compare activity vs logP (even SlogP_VSA2 results) to see what the correlation value is.

Overall the study is worthy of publication. I have attached pdf with other small suggestions and changes. 

Author Response

Reviewer 1

 Dear Reviewer, thank you very much for your attention to our manuscript and for the comments and suggestions to improve the quality of the manuscript. Please, find below our answers to your remarks.

 The is not much justification for the cell line chosen and should be addressed.

- The paragraph in the Introduction on the use of Neuro-2a cell line in our study has been expanded, a new reference has been added (Line 63-72).

Results could be compared to previous work on other cells lines. 

- Our previous work on other cells lines were inserted in Discussion section and compared with present results (Line 353-362).

The SAR study concludes not with positional information about groups but mainly concluding that lipophilic properties are the main determinant of activity.

- As can be seen from the data in Table 1 and Figure 9, highly and moderately toxic compounds contain hydrophobic S- and O- monosaccharide residues and hydrophobic functional groups at C2, C3 and C6, C7 of 5,8-dihydroxy-1,4-naphthoquinone. For the investigated series of compounds, it was found that the replacement of one hydrophobic group at C2 with a polar OH group leads to a loss of cytotoxic activity (compounds U-634 and U-633). To elucidate the role of polar substituents in the construction of the QSAR model, descriptors were chosen that are important for describing the properties of this series of compounds.

I would the authors to compare activity vs logP (even SlogP_VSA2 results) to see what the correlation value is.

When constructing the model, the SlogP_VSA2 descriptor was used and the relative importance of this descriptor is shown in the article in Figure 7 (Line 274). At the request of the reviewer, for the resulting QSAR model, we present here the dependence of pEC50 on SlogP_VSA2: (Please, look at the Figure 1 in the attached Cover letter).

Figure: 1. рEC50 dependence on the SlogP_VSA2 descriptor value in the QSAR model for 17 compounds.

As can be seen from the Figure 1, the correlation coefficient pEC50 from SlogP_VSA2 amounts to R2 = 0.2353 that is relatively low.

In this regard, the following paragraph was inserted into the text of the manuscript (Line 281-286): “The other descriptors we used do not play such a significant role and have rather low values of relative importance of descriptors (RID) compared to ASA and vsurf_S …”

Reviewer 2 Report

The manuscript titled "Synthesis, Cytotoxic Activity Evaluation and Quantitative Structure‐Activity Analysis of Substituted 5,8-Dihydroxy-1,4‐Naphthoquinones and their O- and S-Glycoside Derivatives Tested Against Neuro-2a Cancer Cells" by Sergey et al., evaluates here the cytotoxic activity of 6,7-substituted 2,5,8-trihydroxy-1,4-napthoquinones (1,4-NQs) and its synthesised derivatives.

Although, the manuscript here is well planned but lacks in the quality of presentation with many grammatical as well as typological errors. Hence, it is advised to consult a native speaker or go through the manuscript very carefully before resubmission. I am pointing below some of the minor as well as major points that must needs to be addressed before resubmission.

Major points:

  1. Introduction, it is advised to provide more citations in the first paragraph of the introduction.
  2. Figure 2. There are several R groups in the same compound, whereas only R was designated in the variants, Does that it was all R has same substitution. Please name them R1, R2 and R3.....wherever needed...
  3. Figure 5, quality is very poor. Also, A positive control is missing in viability assay. P
  4. How many times viability assay was repeated? Please provide the details of repetition with statistics. 

Minor points: I am just pointing out few example below, but the manuscript need to undergo extensive check.

Line 19: change to urchins,

Line 28: change to sharp increase

Line 52: change to echinoderms and microorganisms [2].

Line 53: change to drug-leads

Author Response

Reviewer 2

 Dear Reviewer, thank you very much for your attention to our manuscript and for the comments and suggestions to improve the quality of the manuscript. Please, find below our answers to your remarks.

 Major points:

  1. Introduction, it is advised to provide more citations in the first paragraph of the introduction.

- Additionally three new citations were inserted in the first paragraph of the introduction. In Reference section the references were renumbered.

  1. Figure 2. There are several R groups in the same compound, whereas only R was designated in the variants, Does that it was all R has same substitution. Please name them R1, R2 and R3.....wherever needed...

- Figure 2. - It’s OK. Compounds 1214 contains only R = Ac4Glc, and  compounds 1517 have substituents R =  Glc. Radical numbering R1, R2, R3, etc. is not necessary.  Structure of the substituents Ac4Glc and Glc also presented in Figure 2.

- Table 1. Figures naphthazarin core and sugar substituents reformatted and increase, added positive control value and statistic data.

  1. Figure 5, quality is very poor. Also, A positive control is missing in viability assay. P

- New Figure 5 of good quality was inserted into manuscript.

- Cladoloside C was used as a reference cytotoxic compound (positive control). This information was included into section 4.3. EC50 value for reference compound was inserted to the Table 1.

  1. How many times viability assay was repeated? Please provide the details of repetition with statistics. 

- The number of repetitions made for experiments was included into section 4.3 in the footnotes of Table 1 and into section 4.3. The EC50 values were expressed in Table 1 as meam±SEM. This information was included into section 4.3. The range of compound concentrations used for EC50 calculations was included into section 4.3

Minor points: I am just pointing out few example below, but the manuscript need to undergo extensive check.

Line 19: change to urchins,

Line 28: change to sharp increase

Line 52: change to echinoderms and microorganisms [2].

Line 53: change to drug-leads

- All these corrections were made in the text. English was improved. A grammar or spelling mistakes presented in the manuscript were fixed.

Reviewer 3 Report

This manuscript describes a nice investigation of cytotoxicity of naphthoquinones and their derivatives against a neural cancer cell line. A QSAR study was also performed to shed light on chemico-physical aspects related to activity of molecules.

Overall, the work is well organized and described with a satisfactory clarity. The synthesis and the experimental data of molecules were described accurately. 

I suggest to make some general improvement, as follows:

1) a revision of language, because some grammar or spelling mistakes are present in the manuscript.

2) the abstract is too complex in my opinion. The authors should avoid to mention all the descriptors used in QSAR, avoiding the use of non standard abbreviations. This part of the article requires the maximum clarity for readers, in my opinion.

3) in the paragraph 2.5, in Table 1 some important data are missing. How many repetitions were made for experiments? Standard deviations must be included in the value of EC50s. What is the range of concentrations used for EC50 calculations? These data must be added in the table or in the footnotes. No reference compound was used in this assay? 

4) Lines 209-213: in part c, low toxic range is 30-90 μM, and not 30 μM < EC50 =  30-90μM

5) Paragraph 3: in my opinion this discussion is too long. I suggest to strenghten the paragraph to very important content, avoiding too long descriptions.

Author Response

Reviewer 3

 Dear Reviewer, thank you very much for your attention to our manuscript and for the comments and suggestions to improve the quality of the manuscript. Please, find below our answers to your remarks.

 I suggest to make some general improvement, as follows:

  • a revision of language, because some grammar or spelling mistakes are present in the manuscript.

- English was improved. A grammar or spelling mistakes presented in the manuscript were fixed.

2) The abstract is too complex in my opinion. The authors should avoid to mention all the descriptors used in QSAR, avoiding the use of non standard abbreviations. This part of the article requires the maximum clarity for readers, in my opinion.

- Non standard abbreviations of QSAR descriptors were removed from the Abstract. This sentence (Line 33-34) was rephrased.

3) In the paragraph 2.5, in Table 1 some important data are missing. How many repetitions were made for experiments? Standard deviations must be included in the value of EC50s. What is the range of concentrations used for EC50 calculations? These data must be added in the table or in the footnotes. No reference compound was used in this assay? 

- The number of repetitions made for experiments was included into section 4.3 in the footnotes of Table 1 and into section 4.3

- The EC50 values were expressed in Table 1 and in the text of manuscript as meam±SEM. This information was included into section 4.3.

- The range of compound concentrations used for EC50 calculations was included into section 4.3

- Cladoloside C was used as a reference cytotoxic compound. This information was included into section 4.3. EC50 value for reference compound was inserted to the Table 1.

4) Lines 209-213: in part c, low toxic range is 30-90 μM, and not 30 μM < EC50 =  30-90μM

- Line 217 “30 μM < EC50 = 30-90μM” was changed to “EC50 = 30-90 μM”.

5) Paragraph 3: in my opinion this discussion is too long. I suggest to strenghten the paragraph to very important content, avoiding too long descriptions.

- Discussion section was reduced according to recommendation (Line 363-367 and 387-394)

Round 2

Reviewer 2 Report

The manuscript titled “Synthesis, Cytotoxic Activity Evaluation and Quantitative Structure‐Activity Analysis of Substituted 5,8-Dihydroxy-1,4‐Naphthoquinones and their O- and S-Glycoside Derivatives Tested Against Neuro-2a Cancer Cells” by Sergey et al., has improved substantially after careful revision by the authors. Authors have also provided point wise response to the questions arose. 

Hence, i recommend the manuscript to be considered for publication in its current form.